# Electron paramagnetic resonance as a tool to determine the sodium charge storage mechanism of hard carbon

Bin Wang[1,2], Jack R. Fitzpatrick [1,2,3], Adam Brookfield[4], Alistair J. Fielding [5], Emily Reynolds [6], Jake Entwistle[1], Jincheng Tong[7], Ben F. Spencer [8], Sara Baldock [1], Katherine Hunter[9], Christopher M. Kavanagh[9] & Nuria Tapia-Ruiz [1,2,3] ✉

Hard carbon is a promising negative electrode material for rechargeable sodium-ion batteries due to the ready availability of their precursors and high reversible charge storage. The reaction mechanisms that drive the sodiation properties in hard carbons and subsequent electrochemical performance are strictly linked to the characteristic slope and plateau regions observed in the voltage profile of these materials. This work shows that electron paramagnetic resonance (EPR) spectroscopy is a powerful and fast diagnostic tool to predict the extent of the charge stored in the slope and plateau regions during galvanostatic tests in hard carbon materials. EPR lineshape simulation and temperature-dependent measurements help to separate the nature of the spins in mechanochemically modified hard carbon materials synthesised at different temperatures. This proves relationships between structure modification and electrochemical signatures in the galvanostatic curves to obtain information on their sodium storage mechanism. Furthermore, through ex situ EPR studies we study the evolution of these EPR signals at different states of charge to further elucidate the storage mechanisms in these carbons. Finally, we discuss the interrelationship between EPR spectroscopy data of the hard carbon samples studied and their corresponding charging storage mechanism.

Lithium-ion batteries (LIBs) have made great contributions to clean and renewable energy storage and are nowadays indispensable technology in vehicular applications to achieve zero net emissions[1,2]. However, scarcity, unevenly distributed lithium sources and subsequent increased and unpredictable Li costs are becoming the bottleneck for their future deployment[3,4]. Emerging rechargeable battery technologies such as sodium-ion batteries have attracted much attention in recent years due to the low cost and large natural abundance of sodium sources[3,4]. Extensive research efforts have been made in developing suitable electrode materials, particularly negative

[1]Department of Chemistry, Lancaster University, Lancaster LA1 4YB, UK. [2]The Faraday Institution, Harwell Science and Innovation Campus, Quad One, Didcot OX11 0RA, UK. [3]Department of Chemistry, Molecular Sciences Research Hub, White City Campus, Imperial College London, London W12 0BZ, UK. [4]The National Research Facility for Electron Paramagnetic Resonance, Photon Science Institute, University of Manchester, Oxford Road, Manchester M13 9PL, UK. [5]Centre for Natural Products Discovery, School of Pharmacy and Biomolecular Sciences, Liverpool John Moore University, Byrom Street, Liverpool L3 3AF, UK. [6]ISIS Neutron and Muon Spallation Source, STFC Rutherford Appleton Laboratory, Harwell, Oxford OX11 0QX, UK. [7]Department of Chemistry, University of Manchester, Oxford Road, Manchester M13 9PL, UK. [8]Department of Materials, University of Manchester, Oxford Road, Manchester M13 9PL, UK. [9]Deregallera Ltd, Unit 2 De Clare Court, Pontygwindy Industrial Estate, Caerphilly, Wales CF83 3HU, UK. ✉e-mail: n.tapia-ruiz@imperial.ac.uk

electrodes, where it was found that graphite, the state-of-the-art Li-ion battery anode, was not suited for Na-ion batteries when combined with commercial ester-based electrolytes, due to its low specific capacity of ca. 20 mAh g$^{-1}$ [5].

Non-graphitic hard carbons (HC), formed by randomly-oriented and curved graphene sheets with expanded interlayer distance (3.6–4 Å), and with the possibility of being derived from biosources, are promising anode candidates for Na-ion batteries, showing typical reversible capacities of ca. 300 mAh g$^{-1}$ (see refs. [6–19]). The specific capacity of the HC anode comes from two distinct regions in the galvanostatic profile: the slope capacity at high voltages (at V > 0.1 V vs. Na$^+$/Na) and the plateau capacity (at V < 0.1 V vs. Na$^+$/Na). Currently, the sodium storage mechanism of hard carbon materials is still controversial and there are four prevailing mechanistic models, including the insertion–adsorption model[11,19,20], adsorption–intercalation model[21–23], three-stage model[15,16], and adsorption–filling model[6,24,25]. These electrochemical features are associated with three different Na$^+$ ion storage mechanisms in HC: (1) Na$^+$ ion adsorption at edge sites and defects; (2) Na$^+$ intercalation into graphene–graphene interlayer spacing; and (3) Na$^+$ ion insertion into nanopores. The correlation between these charge storage processes, and their electrochemical signature has been a subject of debate in recent years.

Typically, the slope capacity has been associated with the presence of surface groups and defects in HC[15,26], although recent mechanistic studies on hydrothermal carbons suggested that sodium intercalation also occurred at the slope region[17]. On the other hand, the plateau capacity is controlled by the number and accessibility of the closed pores and the prevalence of ordered graphitic-like domains within the HCs and the efficiency of Na$^+$ ion diffusion into these domains[8,25,27–31]. The initial (micro)structural features of HCs determine their electrochemical behaviour and several mechanistic studies involving various advanced characterisation techniques have been adopted to shed light on the structure–performance relationship in these materials[23,32–34].

Electron paramagnetic resonance (EPR) spectroscopy is a fast, sensitive, and powerful tool to investigate the structure and chemical changes in carbonaceous materials when subjected to different physico-chemical processes by probing their unpaired electron density. For example, structural transformations of organic precursors during thermal pyrolysis treatments have been identified by monitoring changes in the EPR signal, which were attributed to paramagnetic centres assigned to aromatic π radicals[35–42]. Furthermore, ex situ EPR has been used to elucidate charge compensation mechanisms on post-mortem carbonaceous materials[12,43,44]. Na$^+$ ions reacting during discharge (at V > 0.1 V) with hydrogen and/or dangling bonds with unpaired electrons at the terminations of the graphene layer in disordered carbons formed C–Na covalent bonds, which decreased the intensity of the localised Curie-type EPR signal as a function of the voltage[43,44]. By contrast, in ordered carbonaceous structures such as graphite, the Pauli-type EPR signal increased in intensity upon Li$^+$ ion intercalation, due to spin–orbit interactions between intercalated ions and graphite crystallites[43,45]. Moreover, EPR has been used to confirm the formation of Na metallic or quasi-metallic clusters at low voltages, akin to Li metal in analogous systems in LIBs[12,42,46]. Studies conducted by Alcántara et al. draw some correlations between spin density and reversible capacity values obtained from the EPR spectra of HC microbeads annealed at different temperatures, showing that the samples with the lowest spin density had the highest reversible capacities[43]. To date, no studies have elucidated the relationship between the EPR spectra of pristine HC materials and their corresponding slope and plateau capacities, which are vital to understanding their electrochemical behaviour. This will be the subject of this work, the prediction of electrochemical responses of different HC materials through the use of EPR spectroscopy.

In this work, we conducted a systematic study using two hard carbon materials synthesised from sustainable biowaste precursors at different carbonisation temperatures. As-received hard carbons were then mechanochemically treated to modify their (micro)structure properties, such as defects, oxygen functionality and open and closed porosity. Lineshape simulation on the EPR spectra at room temperature and 10 K and temperature-dependent EPR studies on these materials, together with their subsequent post-mortem analysis at different discharge states of charge in Na half-cells, was used to correlate their structural and electrochemical properties, to elucidate their charge storage mechanism.

## Results and discussion

### Structural characterisation of pristine and ball-milled hard carbon samples

Two hard carbon materials prepared from the carbonisation of biowaste at 700 °C and 1000 °C, here denoted as HC700 and HC1000, were ball-milled at 400 rpm for 2 h and 5 h. Pristine HC700 and HC1000 consisted of agglomerated particles with irregular morphology and a wide size range, from hundreds of nanometres to a few microns (Supplementary Figs. S1 and S2), indicating that the reaction temperature had little effect on the particle microstructure, as reported in previous work[47]. Ball-milling of both samples further reduced the particle dimensions, leading to a narrower particle size distribution, which was evident after 5 h of treatment.

Powder X-ray diffraction (XRD) data of pristine and ball-milled HC700 and HC1000 samples showed two main broad reflections at ca. 24° and 44° 2θ, which are characteristic of the (002) and (100) Bragg peaks of graphite (Supplementary Fig. S3)[12]. The average interlayer distance ($d_{002}$) in both pristine HC700 and HC1000 materials was calculated as 0.38 nm and 0.37 nm, respectively (Table 1). These values are larger than those of graphite (0.33 nm)[48]. The difference in $d_{002}$ values between both samples is expected to be small given the carbonisation temperatures used, as previous findings have shown that the $d_{002}$ values of HCs only begin to significantly decrease above synthesis temperatures of 1400 °C[28]. Ball-milling these samples resulted in almost negligible changes in the $d_{002}$ interlayer spacing (Supplementary Fig. S4 and Table 1) and subtle changes in crystallinity with

**Table 1 | Summary of selected structural parameters of pristine and ball-milled HC700 and HC1000 samples obtained from XRD, XPS, Raman, SAXS and BET data**

| Sample | XRD | XPS | | | | Raman | SAXS | | | | BET |
|---|---|---|---|---|---|---|---|---|---|---|---|
| | $d_{002}$/nm | C/% | O/% | N/% | C/O | $I_{D3}/I_G$ | $a_1$/arb.units | A/arb.units | B'/arb.units | B''/arb.units | S/m$^2$ g$^{-1}$ |
| HC700 | 0.38 | 92.5 | 3.6 | 3.9 | 25.6 | 1.01 | 3.19 | 3.2E-6 | 1.0E-4 | 9.9E-6 | 326.9 |
| HC700-400-2h | 0.37 | 90.0 | 5.6 | 4.5 | 16.2 | 1.05 | 3.56 | 4.1E-6 | 8.8E-5 | 6.9E-7 | 211.6 |
| HC700-400-5h | 0.37 | 88.3 | 7.8 | 4.0 | 11.4 | 1.24 | 4.24 | 1.7E-5 | 1.2E-5 | 6.7E-8 | 107.2 |
| HC1000 | 0.37 | 96.3 | 1.7 | 2.0 | 57 | 0.65 | 3.95 | 8.0E-6 | 2.1E-4 | 1.4E-5 | 27.1 |
| HC1000-400-2h | 0.37 | 93.5 | 4.8 | 1.7 | 19.3 | 0.67 | 4.08 | 7.1E-6 | 1.1E-4 | 6.5E-6 | 22.4 |
| HC1000-400-5h | 0.37 | 87.1 | 9.7 | 3.2 | 9 | 0.76 | 5.39 | 1.2E-5 | 9.3E-5 | 3.2E-6 | 203.9 |

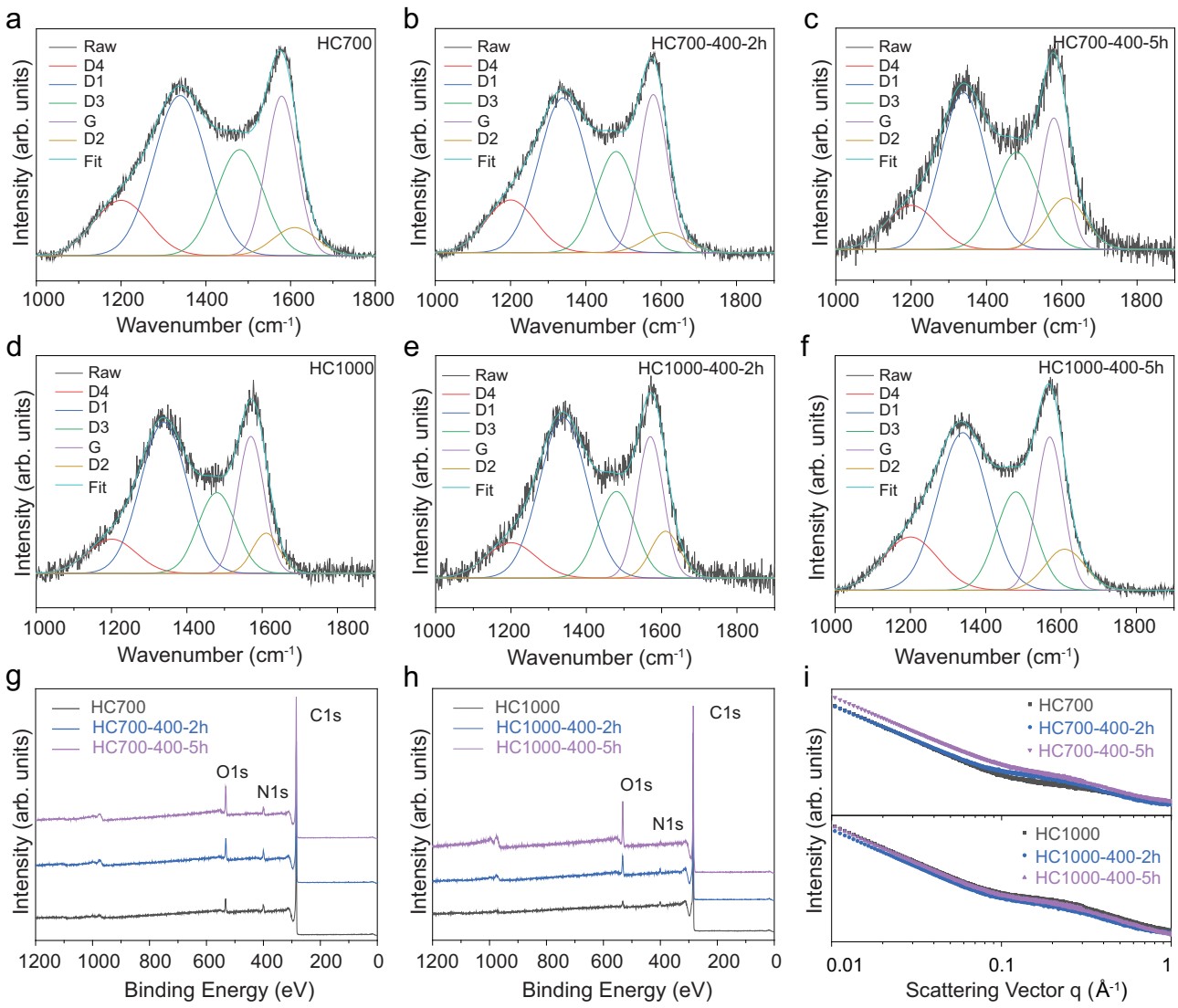

**Fig. 1 | Structural characterisation of pristine and ball-milled HC samples.** Raman spectra of **a–c** HC700 and **d–f** HC1000 samples; Survey XPS spectra of **g** HC700 and **h** HC1000 samples; **i** SAXS data of HC700 and HC1000 samples. Source data are provided as a Source Data File.

milling conditions (as reflected by the FWHM calculated values for these samples, Supplementary Table S1), as shown in earlier reports[32].

Evidence of defects in pristine and ball-milled samples was analysed with Raman spectroscopy, which has been shown to probe < 50 nm in depth in graphite[49]. Raman spectra in the 1000–1700 cm⁻¹ range were deconvoluted into five Lorentzian peaks (from left to right) (Fig. 1a–f): D4 (≈ 1150–1200 cm⁻¹, $sp^2$–$sp^3$ hybrid structure or C-C/C = C stretching vibrations at the graphene terminated edge; $A_{1g}$ symmetry); D1 (≈ 1340 cm⁻¹, defect-induced graphite; $A_{1g}$ symmetry); D3 (≈ 1500–1550 cm⁻¹, short-range $sp^3$ carbon in amorphous carbon); G (≈ 1575 cm⁻¹, stretching vibration of pairs of $sp^2$ carbon atoms; $E_{2g}$ symmetry); and D2 (≈ 1620 cm⁻¹, surface graphene layer of the graphite crystal; $E_{2g}$ symmetry)[50–54]. The degree of graphitisation in the samples was determined by calculating the $I_{D1}/I_G$ peak area ratio[48,50], while the $I_{D3}/I_G$ peak area ratio provided an indication of the oxygen-containing functional groups in the samples[51]. $I_{D1}/I_G$ and $I_{D3}/I_G$ values are shown in Fig. 2a and Table 1, respectively. The higher carbonisation temperature used to produce HC1000 led to lower $I_{D1}/I_G$ and $I_{D3}/I_G$ ratios compared to the HC700 samples, showing a higher degree of in-plane ordering and a lesser presence of oxygen-containing terminal groups, respectively. This behaviour was analogous to that observed in other HC materials synthesised from biowaste precursors[17,48]. Upon extending

the milling time, $I_{D1}/I_G$ and $I_{D3}/I_G$ ratios increased in both cases due to higher defect site concentration and the increase of carbon-oxygen linkages[26]. Similarly, $L_a$ values calculated using Eq. (2) indicated a reduction of the graphitic domains upon milling, as expected from particle fragmentation caused by this treatment (Fig. 2b).

A higher carbonisation temperature reduced successfully the $sp^3$ hybrid carbon atoms at the defect sites of HC and C–O linkages[55], as confirmed by high-resolution C 1$s$ and O 1$s$ X-ray photoelectron spectroscopy (XPS) data (Fig. 1g, h, Supplementary Figs. S5–7 and Supplementary Table S2). XPS data showed a two-fold increase in the carbon-to-oxygen (C/O) atomic ratio with increased temperature (C/O = 25.6; HC700) compared to C/O = 57; HC1000) (Table 1). Ball-milling of both samples causes the breakage of the carbon structure and posterior oxidation of the newly formed carbon dangling bonds, as reflected by the decrease in C/O ratio with ball-milling time (e.g., C/O = 9; HC1000-400-5h, Table 1)[32]. Further evidence suggesting an increase in the oxygen functionalities with ball-milling was obtained from the high-resolution C 1$s$ XPS spectra, which showed an increase in relative intensity for the C–O (286.1 ± 0.2 eV), C = O (287.2 ± 0.2 eV) and O–C = O (288.9 ± 0.2 eV) peaks with respect to the $sp^2$ C (284.5) and π–π*

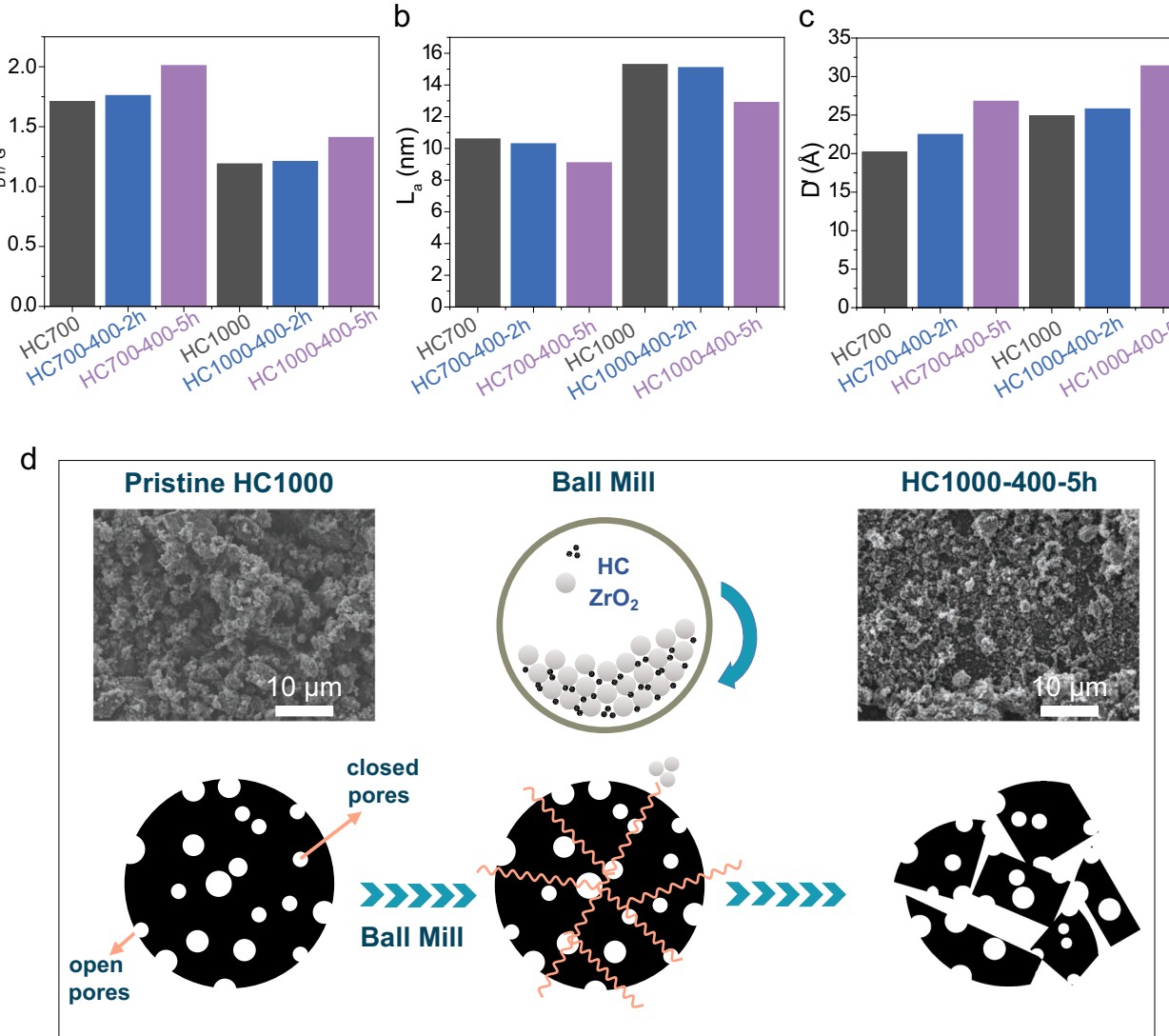

**Fig. 2 | Selected structural parameters obtained from Raman and SAXS data and schematic showing proposed effects of ball-milling on the microstructure of hard carbon. a** $I_{D1}/I_G$ and **b** $L_a$ values calculated from Raman spectroscopy data; and **c** D' values calculated from SAXS data for pristine and ball-milled HC700 and HC1000 samples. **d** Schematic showing the effects on open and closed porosity in HC samples after the ball-milling treatment. Source data are provided as a Source Data File.

(290.9 ± 0.2 eV) peaks after ball-milling (Supplementary Figs. S5 and S6).

Open and closed porosity in HCs were assessed using small-angle X-ray scattering (SAXS), where micropores and mesopores in the bulk can be detected due to differences in scattering density. SAXS patterns showed two domains: a first region in the $0.01 < q < 0.1\,\text{Å}^{-1}$ range, where there is a decrease in scattering intensity ca. 4th power of $q$ from the scattering of the particle surface, followed by a second region with a plateau extending to $q = 0.1–0.3\,\text{Å}^{-1}$, corresponding to the presence of micropores and nanometre-sized voids between $sp^2$ carbon planes (Fig. 1i)[56]. SAXS data were fitted using Eq. (3), assuming previous models described in the literature[10,32,57], and simulation parameters for each sample dataset are listed in Table 1.

The diameter of the closed pores (D') (which is proportional to $a_1$, a size factor related to the radius of a spherical pore, as shown in Eq. (4)), increased with increasing temperature, where pore dimensions increased from 20 Å (HC700) to 25 Å (HC1000), due to the growth and alignment of the graphitic domains (Fig. 2c)[17]. HC700 and HC1000 showed an increase in the diameter of the closed pores with increased ball-milling time which is attributed to the destruction,

opening and coalescence of the pores (Fig. 2d). The destruction of some of the closed porosity with the mechanical treatment is responsible for the observed reduction in the surface area of the closed pores (B' in Eq. (3)) and the relative number of pores (B'' in Eq. (5)) (Table 1).

In both cases, the surface area of the open pores (A in Eq. (3)) increased with ball-milling time (Table 1) due to the breaking and creation of smaller particles, as shown in the SEM images (Supplementary Figs. S1 and S2), promoting the formation of intraparticle voids. These data matched the Brunauer-Emmett-Teller (BET) surface area data from $N_2$ absorption/desorption experiments directed at measuring open porosity for HC1000 (Supplementary Fig. S8 and Table 1). Pristine HC700 showed higher $N_2$ BET surface area than HC1000, which is expected with the closing of the microporosity at high temperatures[58]. The decreased BET surface areas observed upon increasing ball-milling time for the HC700 samples, which showed an opposite trend to the HC1000 samples, could be explained by the presence of smaller open pores, which might be undetected using $N_2$ gas and thus, would require the use of other gases such as $CO_2$ and He[58]. Alternatively, it might be attributed to particle agglomeration

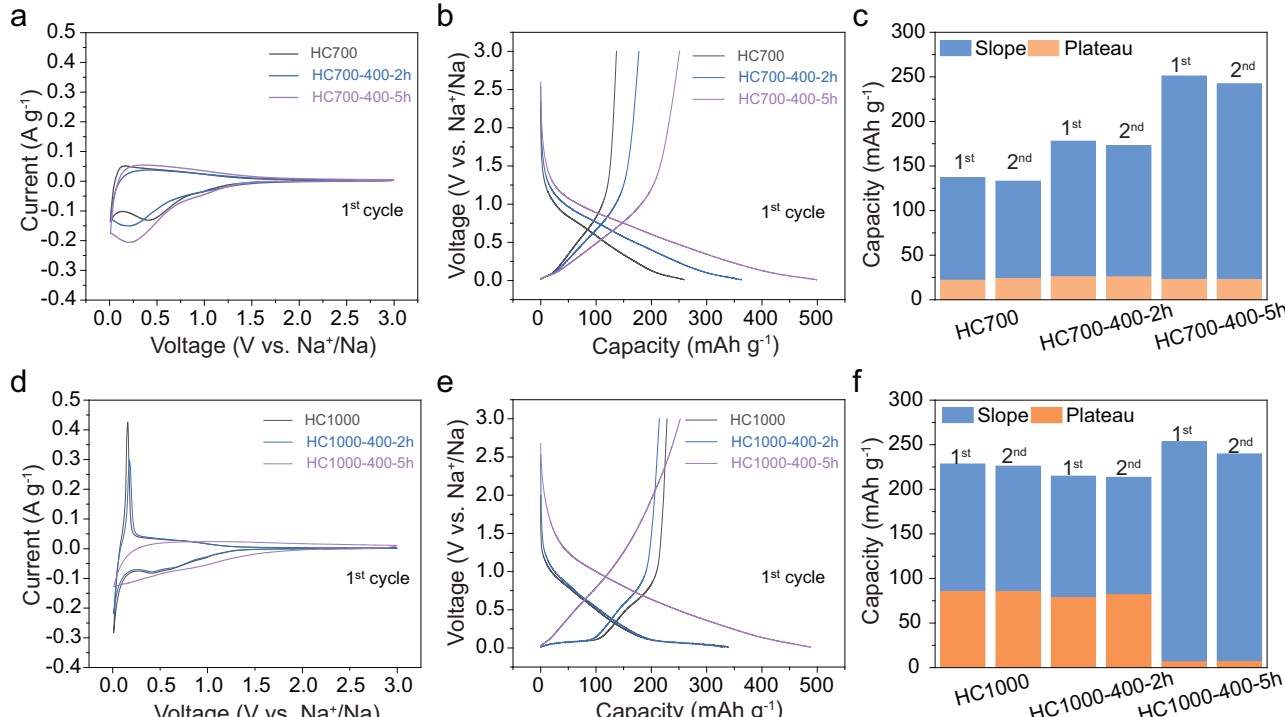

**Fig. 3 | Electrochemical data of pristine and ball-milled HC samples during the first cycle and summary of slope vs. plateau capacity contributions.** Cyclic voltammetry data of **a** HC700 and **d** HC1000 samples; Galvanostatic charge/discharge curves at a current rate of 5 mA g$^{-1}$ of **b** HC700 samples and **e** HC1000 samples; Slope and plateau capacity contribution to the total capacity during the first and second charge processes for **c** HC700 samples and **f** HC1000 samples. All cells were cycled in the voltage range of 3–0.01 V vs Na$^+$/Na. Source data are provided as a Source Data File.

caused by the ball-milling treatment, which might have blocked some of the open pores.

## Electrochemical performance of pristine and ball-milled hard carbon samples

Electrochemical testing of pristine and ball-milled HC700 and HC1000 samples as negative electrodes was conducted in half-cells using Na metal as the counter and reference electrode. Na metal is unstable in carbonate-based electrolytes, resulting in significant polarisation, which has been shown to increase with cycling and current rate[59–61]. The polarisation of Na metal has a significant effect on the observed capacity of hard carbon materials when tested in half-cells. This is because hard carbon materials exhibit a significant proportion of their capacity in a low-voltage (< 0.1 V) plateau region. Thus, any changes in potential of the Na metal due to polarisation result in the lower cut-off voltage (in this case, 0.01 V) being reached prematurely, cutting off the plateau region of the hard carbon and underestimating its capacity. Therefore, we note that the use of Na metal in this work has likely resulted in an underestimation of the specific capacities, relative to their expected performance in a full cell. However, as all hard carbons in this work were electrochemically tested in an identical manner, the observed differences in electrochemical performance presented in the following section are valid, and a comparative assessment between samples can be established. The carbonisation temperature affected the electrochemical behaviour of HC700 and HC1000, as evidenced by the different galvanostatic charge/discharge (GCD) curves (Fig. 3 and Supplementary Fig. S9). HC1000 exhibited a first reversible charge capacity (sodium extraction) of 229 mAh g$^{-1}$, with charge contributions from the slope and plateau regions (Fig. 3e, f and Supplementary Table S3), while HC700 showed a lower reversible charge capacity of 137 mAh g$^{-1}$, which was slope-

dominated (Fig. 3b, c and Supplementary Table S3). During the second charge cycle, the slope:plateau capacity ratio did not change significantly from the first charge cycle for both samples (Fig. 3c, f). To differentiate accurately between slope and plateau capacities, dQ/dV vs. V plots were used to calculate the onset potential where the plateau capacity started (Supplementary Fig. S10 and Supplementary Table S3)[62].

Reports have attributed the sloping region of the galvanostatic curve to charge stored through the adsorption of Na$^+$ ions at the surface of open pores and defects sites of graphene sheets, including the edges, heteroatoms, and vacancies[63–65]; while the low-voltage plateau region has been explained with pore filling and formation of quasi-metallic Na clusters[17]. The texture of the graphitic-like domains will dictate whether there will be a slope or plateau contribution, i.e., Na$^+$ ions insertion between graphene–graphene layers with a wide energy distribution is related to sloping capacity[19], whereas intercalation between graphene layers with phase transition to form Na-graphite-like compounds, corresponds to the plateau capacity[66].

Cyclic voltammetry (CV) data of both pristine samples showed a broad irreversible cathodic peak centred at ≈ 0.5 V attributed to irreversible SEI formation and irreversible trapping of Na$^+$ ions in defects, which was higher in current intensity for HC700, as expected from the lower initial coulombic efficiency (ICE) observed in the GCD curves (Fig. 3a, d and Supplementary Fig. S11)[48]. Moreover, we observed a pair of cathodic and anodic peaks in the low-voltage region (< 0.2 V), sharper and more intense for HC1000, which can be related to the larger plateau capacity observed for this sample. These peaks were ascribed to the insertion of quasi-metallic Na into the closed pores[67].

The ball-milling treatment also revealed significant changes to the electrochemical behaviour of both samples. Focusing first on the HC1000 samples, HC1000-400-2h exhibited a similar capacity to the pristine material, with distinct slope and plateau regions. HC1000-400-5h showed a vastly different electrochemical behaviour to both

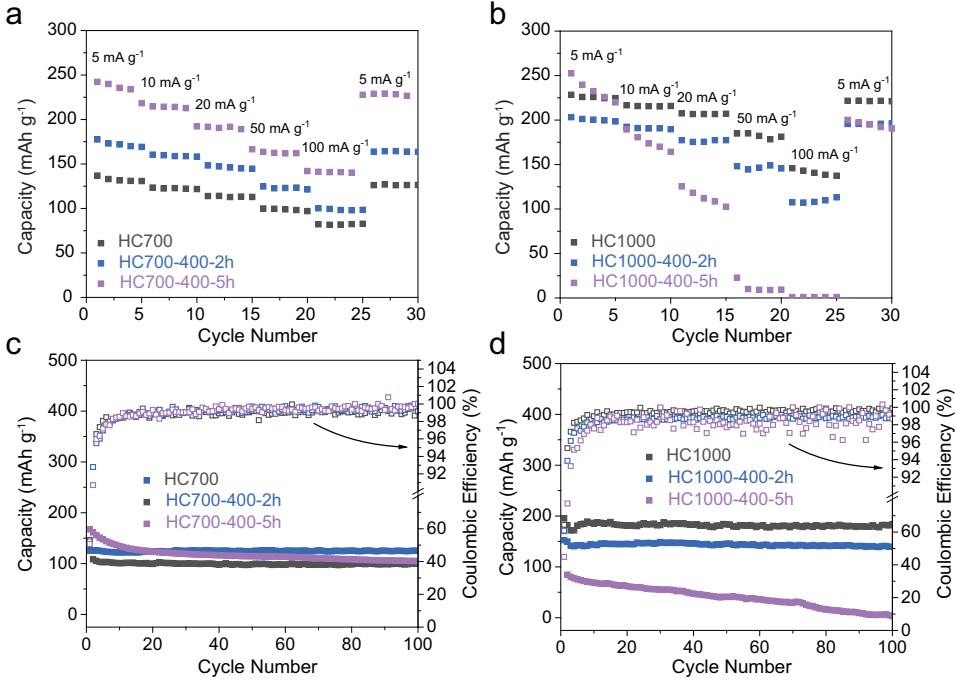

**Fig. 4 | Long-term electrochemical performance and rate capability of pristine and ball-milled HC samples.** Rate capability tests at 5,10, 20, 50, 100 and 5 mA g$^{-1}$ of **a** HC700 and **b** HC1000 pristine and ball-milled samples. Each current was applied during five cycles. Cycling performance at a current rate of 50 mA g$^{-1}$ of **c** HC700 and **d** HC1000 pristine and ball-milled samples and their respective coulombic efficiency values. All cells were cycled in the voltage range of 3–0.01 V vs Na$^+$/Na. Source data are provided as a Source Data File.

the pristine and the HC1000-400-2h samples, achieving a first cycle reversible capacity of 254 mAh g$^{-1}$, which was mostly slope dominant (Fig. 3f and Supplementary Table S3). CV data also reflected these changes, showing the disappearance of the redox peaks at < 0.2 V, resembling the shape of a capacitor-like material after 5 h of milling. HC700 samples followed a similar trend as HC1000 in terms of initial charge capacity and ICE with increased ball-milling time (Supplementary Table S3). In all cases, the specific capacity was dominated by the sloping voltage region, and its contribution increased with ball-milling time (Supplementary Table S3). Overall, we observed a higher insertion potential with increased ball-milling time for HC700 and HC1000 samples, which we attributed to the increased number of surface oxygen functional groups and carbon defects[68,69].

Rate capability and long-term cycling stability studies of the HC samples were also conducted (Fig. 4, Supplementary Fig. S12 and Supplementary Table S4). Generally, for both sets of HCs, poorer capacity retention with increasing ball-milling time was observed, suggesting that overall, storage of Na$^+$ ions through the slope is less reversible than through the plateau[65]. This was especially apparent for the HC1000 series with HC1000-400-5h (Fig. 4d), which achieved capacity retention of 8% after 100 cycles compared to 62% for HC700-400-5h at a constant current rate of 50 mA g$^{-1}$. It is worth noting that HC700-400-2h did not follow this trend as it performed better in terms of reversible capacity and capacity retention vs. pristine HC700.

The rate performance, which was characterised at the current rates from 5 to 100 mA g$^{-1}$, showed opposite electrochemical behaviour with increasing ball-milling time for both sample series, whereby HC1000 samples showed decreasing rate capability while HC700 showed increasing rate capability (Fig. 4a, b). Thus, data suggested that changes in the rate capability with ball-milling were more driven by the chosen pristine HC than the applied ball-milling procedure.

Combining our previous structural and electrochemical analysis, we can establish that ball-milling destroys the amount/size of the ordered graphitic regions within the HC structure, as evidenced by the decrease in crystallinity (XRD) and the increase in defect concentration (Raman spectroscopy). This resulted in a decreased particle size (SEM) and a subsequent increase in open porosity (BET surface area and A parameter from SAXS), oxygen content at the surface of the HC particles (XPS, Raman spectroscopy), and diameter of the close pores (D' parameter in SAXS), attributed to pore opening, followed by pore coalescence and subsequent formation of new pores. These structural changes were expressed electrochemically by a decrease in ICE (Supplementary Table S3), due to increased SEI formation, and increased dominance of the sloping capacity, attributed to increased Na$^+$ adsorption at the surface due to the presence of functional groups and defects. The general decrease in capacity retention observed with ball-milling time suggested that Na$^+$ adsorption at these surface sites, which is promoted by ball-milling, is not as reversible over a long-time scale compared to the storage processes occurring in the plateau region, supporting previous literature reports[70].

## Electron paramagnetic resonance studies of pristine and ball-milled hard carbon samples

EPR was used to further study the structural transformations occurring in HC materials during heating and ball-milling treatment to further clarify their sodiation mechanism. Figure 5 compares the EPR spectra of pristine and ball-milled HC700 and HC1000 samples at 10 K, showing the changes in intensity and linewidth of the EPR lines as a function of heat treatment and ball-milling conditions. Supplementary Table S5 shows a summary of g-tensor and peak-to-peak linewidth values ($\Delta H_{pp}$) obtained from the corresponding EPR spectra in Fig. 5. For reference, EPR spectra at room temperature were also collected and are shown in Supplementary Fig. S13 and relevant information is shown in Supplementary Table S6.

All the EPR signals are observed at resonance fields which are typical for carbonaceous radicals and conduction electrons in graphite-like domains. Therefore, we can exclude any contribution

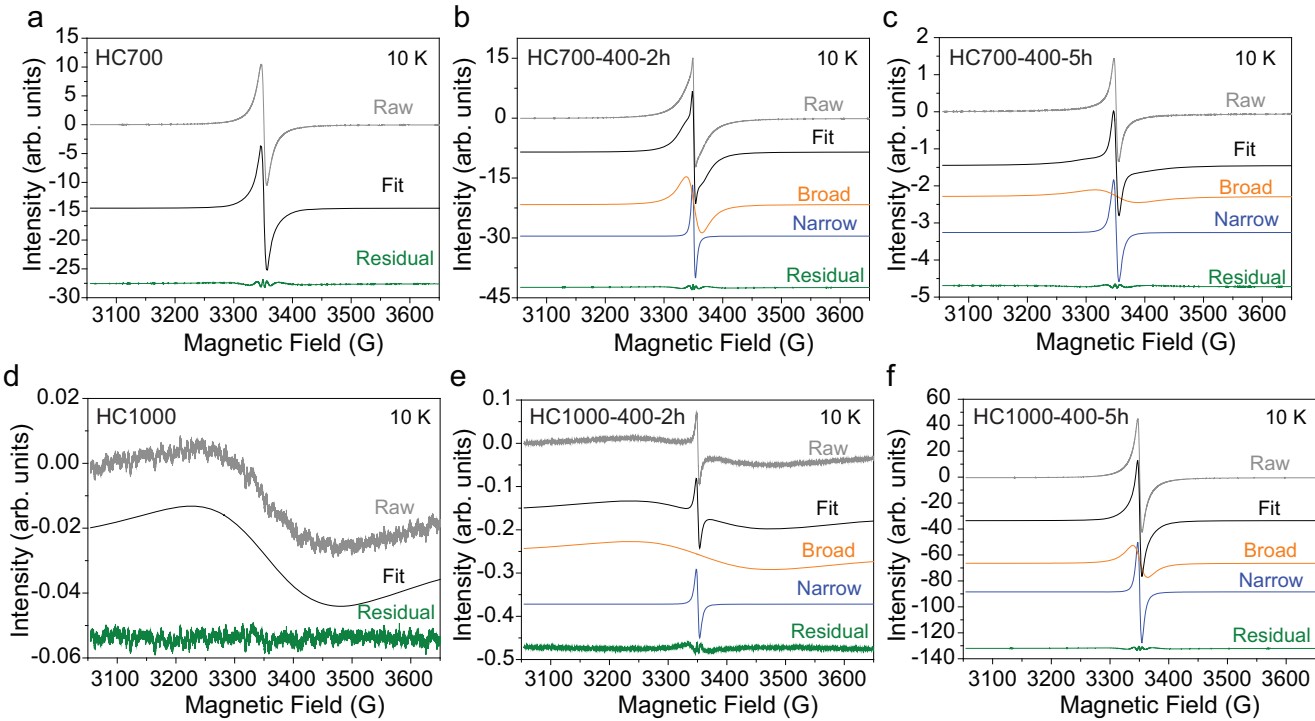

**Fig. 5 | EPR data showing broad and narrow fitting components at 10 K of the HC pristine and ball-milled samples.** EPR characterisation of **a–c** HC700 and **d–f** HC1000 pristine and ball-milled materials at 10 K, using a mass loading of 1 mg. The broad signal (orange) is related to the initial extended aromatic structure in the ball-milled samples, while the narrow signal (blue) appeared after ball-milling and is related to new defects created during the ball-milling process. Source data are provided as a Source Data File.

from paramagnetic species arising from impurities from the ball-milling reactions, which would be characterised by different g-tensor values. Either one or two signals were observed in the EPR spectra and were fitted to either Lorentzian or Dysonian lineshapes, which are in turn, associated with different electron mobility within the samples. Information related to the spectral fitting can be found in the Methods section.

The synthesis temperature had a profound effect on the EPR spectra seen in both pristine samples at room temperature and 10 K. Earlier works on the dependence of the EPR signal with the synthesis temperature of various carbonaceous materials have been reported[38,71,72], showing that the carbonisation process forms localised spins while the graphitisation process generates mobile charge carriers[73,74]. In general, localised spins obey the Curie–Weiss law, whereby the spin susceptibility (which is proportional to the double integration (DI) value of the mass-normalised EPR signal) decreases with increasing temperature; and delocalised electrons will show Pauli behaviour, where the spin susceptibility is independent of the temperature.

EPR spectrum of pristine HC700 at 10 K showed an intense and sharp signal with a Lorentzian lineshape centred at $g = 2.0024$ and $\Delta H_{PP} = 9.5$ G (Fig. 5a) that obeyed the Curie–Weiss law, as expected from the presence of localised spin centres (Fig. 6a)[38]. EPR parameters of the signal are consistent with localised carbon-centred radicals, which can be attributed to a defective carbon structure, such as dangling bonds with terminating oxygen/nitrogen groups (as observed in the XPS data) in the edge of the graphene sheets and/or on the surface of open micropores and vacancies within the graphene sheets[75,76].

Through post-mortem EPR studies at 10 K, we studied the evolution of this resonance during the discharge process at 0.5 and 0.02 V using a constant current of 5 mA g$^{-1}$. Such a low current was chosen to minimise the polarisation of the Na metal counter/reference, subsequently reducing its impact on the electrochemical performance of the

chosen hard carbon. Post-mortem EPR spectra are shown in Fig. 7a, b. The EPR signals observed during discharge to 0.5 V and 0.02 V were similar to the initially observed narrow signal fitted to a Lorentzian lineshape in the pristine material, with $g$ values close to 2.003 and Curie–Weiss type behaviour, indicating the nature of localised spins. A progressive line narrowing occurred when decreasing the potential (from 9.48 G for pristine HC700 to 8 G at 0.5 V, and 6.4 G at 0.02 V), in agreement with previously reported works[44]. The changes in the line-width of the HC700 at different potentials may be explained by alterations in the environment of the free radicals related to the differences in the binding energy of defects/functional groups with Na$^+$ ions[68,69]. The chemical process reflected in the EPR signals is directly echoed in the electrochemical curves with the presence of a slope, which has been previously explained with the sodiation of the dangling bonds over a range of different energies[43,44]. The localised free radicals in pristine HC700 act as electron acceptors towards Na$^+$, forming covalent bonds such as C-Na or O-Na bonds. As the potential decreases, more of these bonds form, leading to a continuous decrease or even disappearance of original localised radicals in disordered carbons, as previous works have reported[36]. The reminiscent EPR signal observed in the spectra after discharge to 0.02 V may be explained by residual bonds which have not reacted with Na. Overall, it was not possible to provide a quantitative measurement of the spins in this study due to difficulties in preserving the electrodes intact during sample preparation. Furthermore, ex situ EPR studies on the two ball-milled HC700 samples after discharge to 0.02 V showed similar EPR signals (Supplementary Fig. S14 and Supplementary Table S7), further revealing an analogous reaction mechanism to HC700.

We did not observe an EPR signal related to the presence of metallic or quasi-metallic Na at 0.02 V which might be related to the insertion of Na into the closed pores of HC or Na plating, despite the observance of a small voltage plateau in the first discharge, which has been attributed to these processes (Fig. 3e)[17,77]. Previous ex situ EPR

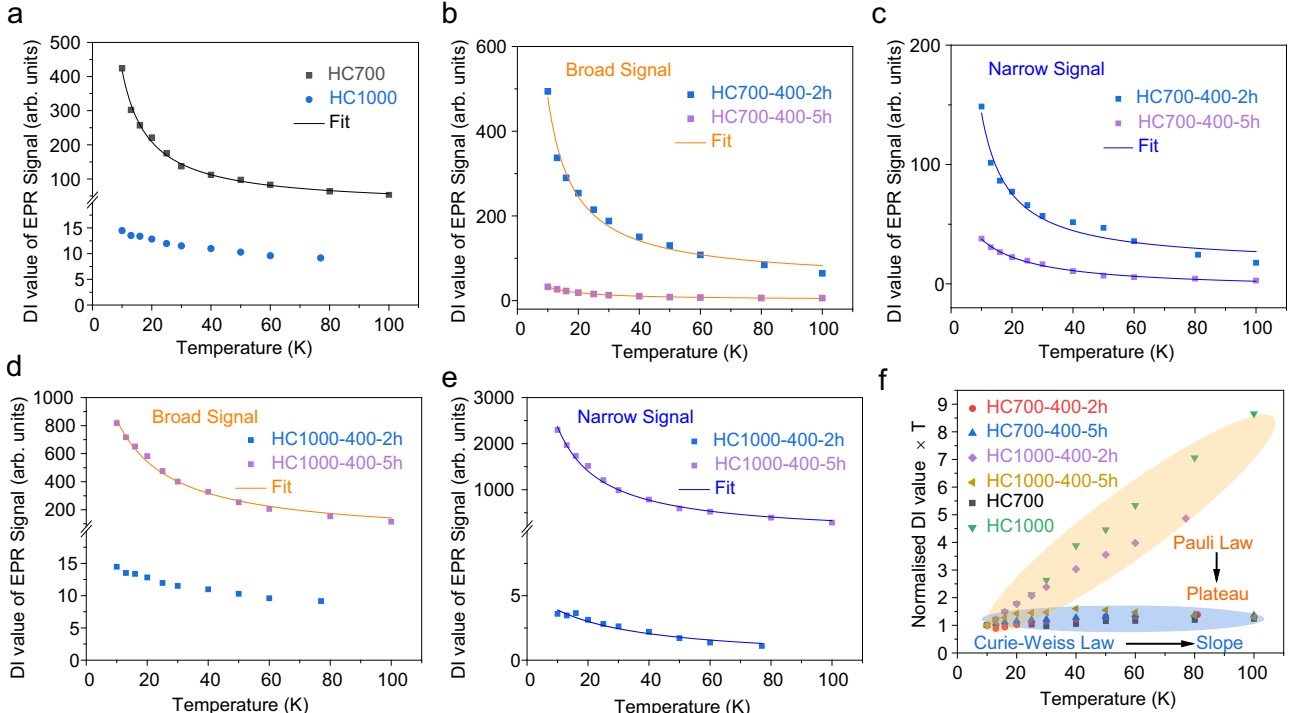

**Fig. 6 | Temperature dependence of EPR signals for pristine and ball-milled HC samples. a** pristine material, **b**, **d** broad and **c**, **e** narrow signal contributions obtained by lineshape simulation for ball-milled HC700 and HC1000 samples. The DI value of the EPR signal is proportional to the spin density and is calculated from the absolute area value of the EPR signal. The solid line in the spectra shows the Curie−Weiss fitting results. **f** Normalised DI value (refers to the spin susceptibility) × T vs. T of the simulated signals of HC700 and HC100 pristine materials and broad signals of the ball-milled samples shown in Fig. 6a, b and d. A mass of 1 mg of sample was used in the measurements. Source data are provided as a Source Data File.

studies have explained the presence of metallic or quasi-metallic Na with the presence of sharp resonances in the spectra. Nevertheless, these signals were observed in HCs synthesised at higher temperatures than 700 °C and thus, with significantly different electrochemical signatures than HC700, including longer plateaus > 100 mAh g$^{-1}$[112,78]. Furthermore, data were collected at different electrochemical conditions to those reported in this work, including the over-discharge of the electrode or data collected during the second discharge cycle. Alternatively, it is possible that this signal is weak and overlaps with the narrow signal attributed to the free radicals and therefore it is not possible to overrule this process.

Increasing the temperature from 700 °C to 1000 °C led to a drastic reduction of the EPR signal intensity by a factor of 8, which we attributed to the start of the graphitisation process and subsequent recombination of free radicals and removal of defects. HC1000 showed a broad signal with an asymmetric (Dysonian) lineshape (A/B = 1.07) (g = 2.0044 and $\Delta H_{pp} \approx 150$ G) at 10 K (Fig. 5d) and paramagnetic Pauli behaviour was observed (Fig. 6a)[74]. This signal is attributed to the existence of mobile electrons from intrinsic defects from basal planes in the carbon structure, on which electrons can move without limitation[73,74]. Therefore, it is plausible to relate this signal to storage processes where Na$^+$ ions might interact directly or indirectly with these basal planes (i.e., through Na$^+$ intercalation between graphitic planes or pore-filling insertion[17,66]). Thus, we might associate this signal to the plateau observed in the galvanostatic curves for this sample. It is worth noting that the broadening of the EPR signal did not arise from the presence of O$_2$ molecules adsorbing to the open pores of the HC sample, as seen in other works[36,41], since data were collected under vacuum.

Through post-mortem EPR analysis, we also studied the evolution of this resonance at different states of charge. Upon discharge to 0.5 V, the initial EPR signal of HC1000 with Pauli behaviour disappeared, and two signals with g values close to 2.004 with different linewidths

($\Delta H_{pp}$ = 3 G and 57 G) and displaying Curie−Weiss behaviour appeared (Fig. 7b, d), similar to previously reported results[43,78]. The broader signal decreased in linewidth to 32 G and broad/narrow signal ratios increased from 0.28 to 0.38 when the voltage decreased from 0.5 to 0.02 V. Alkaline ion insertion into graphitic layers has been thought to show an increased EPR signal during the discharging process due to the spin−orbit interactions between intercalated Na/Li and the delocalised π-electrons graphite crystallites[43,79,80]. Thus, an explanation for these observations is that the broad component in the discharged HC1000 sample arises from localised spins when Na$^+$ ions are intercalated into the graphitic structure, and the narrow signal is associated to the formation of localised paramagnetic centres after Na adsorbed on the disordered carbon surface[40]. This is further supported by the increased broad/narrow ratio, as mentioned above, and its decreased linewidth feature over the discharging process, as the increased spin concentration at lower voltage causes further linewidth narrowing. Akin to the HC700 ex situ studies, we could not conclusively determine whether Na quasi-metallic related to Na insertion into the closed pores was present in this sample.

A new EPR narrow signal appeared after ball-milling HC700 and HC1000 (Fig. 5b, c, e, f), which increased in intensity with increased ball-milling times. These trends were better observed when lowering the temperature from room temperature to 10 K due to the freezing of the spins at lower temperatures. This behaviour is analogous to that seen in ball-milled graphite and nanographite[81–83] and is attributed to an increase in defects such as vacancies and carbon/oxygen-centred radicals[83–86] due to particle fragmentation and oxidation of the edge structure caused by the mechanochemical treatment. These results agreed with the observed increase in milling time of the I$_{DI}$/I$_G$ ratio calculated from the Raman data and the decrease in the C/O ratio observed in the XPS data for all the samples. The narrow line in all the ball-milled samples was simulated with a single Lorentzian component centred with an average of g ≈ 2.003

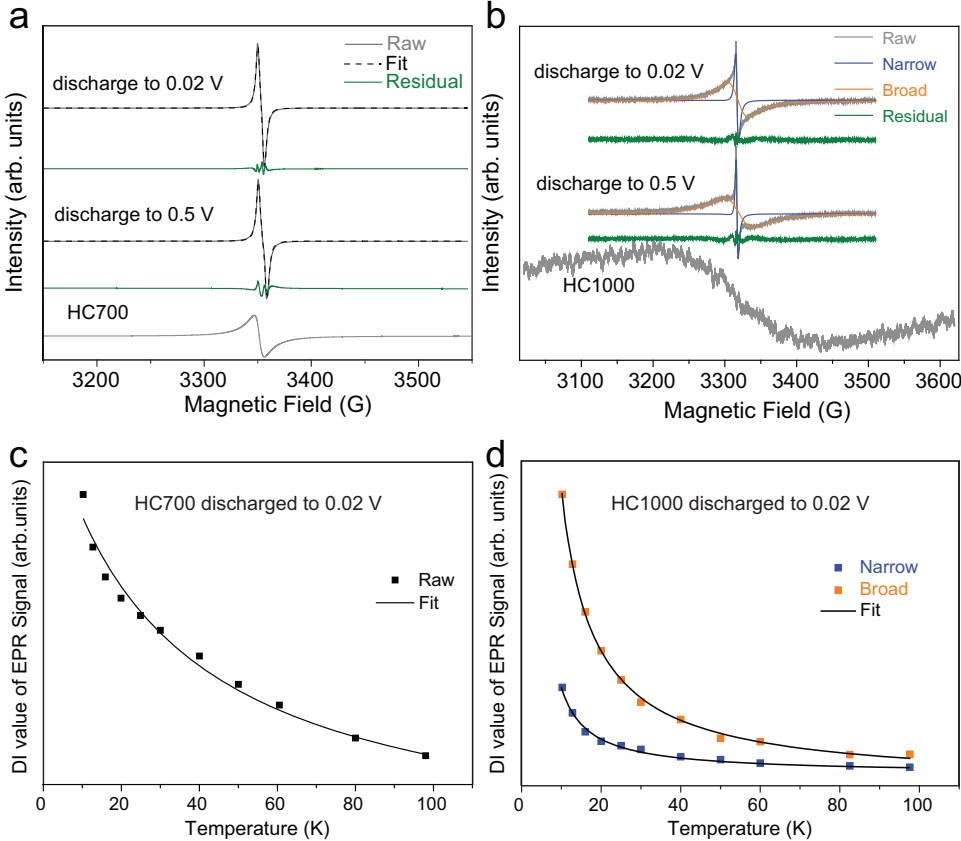

**Fig. 7 | Ex situ EPR data on pristine HC700 and HC1000 samples cycled in Na half-cells.** EPR spectra with simulated signals of pristine HC700 (**a**) and HC1000 (**b**) discharged samples to 0.5 and 0.02 V. Temperature-dependence behaviour of simulated EPR data components in pristine **c** HC700 and **d** HC1000 electrode samples discharged to 0.02 V. Source data are provided as a Source Data File.

and a peak-to-peak linewidth of $\Delta H_{pp} \approx 5–9$. Temperature dependence of the narrow line suggested a Curie−Weiss behaviour from localised electrons (Fig. 6c, e), further verifying the nature of localised spins from dangling bonds at the edge of nano-graphene created during the ball-milling process.

The broad resonances in the ball-milled HC700 samples were simulated using Lorentzian lineshapes and followed the Curie−Weiss law (Figs. 5b, c and 6b). The larger linewidths (Supplementary Table S5) of these resonances compared to the newly generated narrow signals suggested a faster spin relaxation time, as this is inversely proportional to the linewidth[86,87]. Thus, we explain these resonances with the presence of localised π electrons trapped by defects or vacancies and coupled with mobile electrons on extended aromatic domains[86]. The linewidth of the broader signal in all the HC700 samples increased with the ball-milling time (Supplementary Table S5), which might be attributed to the increased intrinsic carbon defects density and the stronger exchange interactions between mobile electrons.

For the ball-milled HC1000 samples, the linewidths of the broader signal exhibited an opposite trend to the HC700 samples, as the linewidth decreased from 150 G (HC1000) to 140 (HC1000-400-2h) and 25 G (HC1000-400-5h) (Fig. 5d–f and Supplementary Table S5). Ball-milling causes a reduction in the range of the extended aromatic structure, which is accompanied by a change in the spin nature in these samples, i.e., from Pauli to Curie−Weiss law behaviour (Fig. 6a, d). This behaviour might be related to carbon structure degradation, which has proved to be detrimental to the electrochemical performance observed in the HC1000-400-5h sample. Ex situ EPR studies also reflected this unusual behaviour, where only one signal was observed after discharge to 0.02 V, suggesting one Na storage process related to the slope capacity behaviour. This contrasts with the two signals

observed in HC1000-400-2h (Supplementary Fig. S14) and HC1000 (Fig. 7), which featured two different Na storage behaviours which might be attributed to both slope and plateau processes.

Figure 6f summarises the double integration of the EPR signal, which is proportional to the spin susceptibility, with temperature for pristine and ball-milled HC700 and HC1000 samples, which allowed us to ascertain trends between the sets of samples with regard to their broad component. For the HC700 samples, which capacity originates from the slope region of the galvanostatic curves, all the broad components of the EPR signal obeyed the Curie−Weiss law, as the signal is almost constant at all the studied temperatures. By contrast, the broad components of the HC1000, and HC1000-400-2h samples, which show plateau-dominated capacities, followed Pauli's law. These samples, however, still had a slope region contributing to the total capacity due to the presence of defects in the carbon structure. The presence of defects is beneficial to increase the slope capacity, although an excessive number of defects may increase the sodium diffusion activation barriers and reduce the plateau capacity[65], as observed in HC1000-400-5h, drifting the nature of the temperature-dependence behaviour of the EPR signal to fulfil the Curie−Weiss law. It is, therefore, not surprising that the capacity of the HC1000-400-5h anode comes solely from the slope region, further corroborating our results.

## Summary
In conclusion, EPR spectroscopy measurements on a series of ball-milled HC samples prepared at 700 °C and 1000 °C (HC700 and HC1000, respectively) were performed to understand the relationship between the carbon structure features from the pristine and ball-milled materials and their corresponding charge storage mechanism conducted in half-cells using Na metal as the counter and reference electrode.

In addition to the effects on structural properties and electrochemical behaviour expected by the use of different carbonisation temperatures to produce HCs, the ball-milling treatment applied to all samples further modifies the structure of these materials, whereby the mechanochemical treatment breaks the carbonaceous framework, reducing the particle size and the number of closed pores and increasing defect concentration, surface area, and oxygen-related defects. These features allowed us to establish correlations with different electrochemical performance metrics when the materials are evaluated as anodes for Na-ion batteries, including ICE, cycling stability, rate performance and capacity contribution to the total capacity (either slope or plateau-dominated capacities). Electrochemical data showed that the increased defective structure led to a decrease in the ICE as well as the fading of the plateau capacity in favour of the slope capacity, caused by an increase in the number of $Na^+$ ion adsorption sites.

Simulation of the EPR resonances of all the samples studied in this work led to the identification of either one or two signals with Lorentzian and Dysonian lineshapes, which corresponded to different electron mobility within the samples and thus, were attributed to different structural features in the hard carbon materials studied. Pristine HC700 and HC1000 samples showed distinctive single resonances in the EPR spectra, whereas a new Lorentzian signal with Curie–Weiss behaviour attributed to an increase in defects (e.g., vacancies and carbon/oxygen-centred radicals) due to particle fragmentation and oxidation of the edge structure was identified in all the ball-milled samples. The identification of these EPR signals led us to predict the electrochemical behaviour of all the pristine and ball-milled materials, i.e., whether the major contributions from the total capacity would arise from the slope or plateau capacity when cycled as anodes in Na half-cells.

EPR spectroscopy revealed a very distinct behaviour among the pristine and ball-milled HC synthesised at different carbonisation temperatures. Pristine and ball-milled samples carbonised at 700 °C, with slope-dominated capacity at V > 0.1 V attributed to $Na^+$ ions absorbed in structure defects, had EPR spectra showing an intense and narrow resonance with a Lorentzian lineshape that obeyed the Curie–Weiss law, attributed to a defective carbon structure, such as dangling bonds with terminating oxygen/nitrogen groups. By contrast, pristine and ball-milled samples carbonised at 1000 °C showed an EPR signal with a broad and asymmetric Dysonian lineshape and Pauli's behaviour, associated with mobile electrons arising from an extended aromatic structure at curved graphene sheets or pore walls. Thus, it was possible to relate this EPR signature with the plateau observed at V < 0.1 V in the galvanostatic curves for this sample. An exception to this behaviour was the sample ball-milled for 5 h, which resembled more the structural and electrochemical properties of the HC700 series and therefore, had a more similar EPR spectra and temperature-dependence susceptibility behaviour to the aforementioned samples.

Ex situ EPR studies of the sodiated HC samples helped to further corroborate the existing differences in charge storage mechanisms between the samples studied, where it was shown that the samples exhibiting plateau-dominant capacity had two different EPR signals after fully discharged, whereas only one signal was observed in the samples with slope dominant capacity.

In conclusion, we demonstrated in this study that EPR is an extremely sensitive, non-destructive, and fast technique to quickly predict the charge storage mechanism of hard carbon anode materials for sodium-ion batteries.

## Methods

### Material synthesis and characterisation

HC powders synthesised from the pyrolysis of biowaste at 700 °C and 1000 °C were obtained from Deregallera. Powdered samples (around 1 g) were ball-milled in a planetary mill (TOB New Energy) at 400 rpm for 2 h and 5 h, using a stainless-steel agate vessel (20 ml) with 10 zirconia balls (5 mm diameter). Samples were named according to the temperature and conditions used in the ball-milling treatment to produce these; for example, HC1000-400-2h indicates that the sample was carbonised at 1000 °C and ball-milled at 400 rpm for 2 h.

SEM images were acquired on a JEOL JSM-7800F operated at 5.0 kV. For the measurement, HC powders were placed onto carbon tabs (G3348N, Agar Scientific).

Powder XRD data were collected on a Rigaku MiniFlex600 at room temperature in Bragg-Brentano geometry with a 0.6 kW Cu-source generator ($K_\alpha = 1.54059$ Å) and a D/teX Ultra detector. All samples were packed on a flat circular plastic insert placed inside a stainless-steel sample holder. To account for differences in sample packing/height, titanium foil was used as a reference for all of the samples. A circular disc of Ti foil was first measured on its own and used as a reference. Subsequent HC powdered samples were packed on top of the Ti foil. Thus, Ti reflections were present in all the XRD data. All the XRD data were shifted so that the most intense Ti reflection was at 38.21° 2θ to match that of the pristine Ti foil reference sample. All samples were scanned from 5° to 70° 2θ. The average interlayer spacing was calculated according to Bragg's law:

$$2d_{002} \sin\theta = n\lambda \tag{1}$$

where $d_{002}$ is the average interlayer spacing, $\theta$ is the diffraction angle of the (002) reflection, $\lambda$ is the wavelength of the X-ray beam (Cu $K_\alpha$ radiation, 0.154059 nm) and n is the order of the reflection. To find accurately the position of the (002) reflection, a Gaussian peak was fitted to the (002) reflection and an asymmetric least squares smoothing baseline was used using the Origin software[88].

Raman spectra were obtained using a Renishaw inVia Confocal Raman microscope with a 523 nm excitation laser ($\lambda_{laser}$) with a power of 0.5 mW at 10% intensity with a 100× objective lens. Spectra acquisition was set to 1 min with five accumulations. The a-axis crystallite size ($L_a$) of the HC samples was calculated according to the following equation[89]:

$$L_a = \left(2.4 \times 10^{-10}\right)\lambda_{laser}^4 \left(\frac{I_{D1}}{I_G}\right)^{-1} \tag{2}$$

where $I_{D1}/I_G$ corresponds to the degree of sample graphitisation.

XPS was performed using a Kratos Axis Ultra DLD spectrometer with a monochromate Al $K_\alpha$ X-ray source ($E = 1486.6$ eV, 10 mA emission). The analysis area was 300 μm² and charge neutralisation was used for all analysis. The spectra were corrected by shifting the peaks to the C $sp^2$ spectral component in the C 1s spectra to 284.5 eV. All data were recorded at a base pressure of below $9 \times 10^{-9}$ Torr and 294 K. XPS spectra were fitted using the Casa XPS software[90].

SAXS measurements were collected on a XENOCS Nano-inXider with a microfocus sealed Cu tube source and two Dectris Pilatus detectors for SAXS. Powdered samples were loaded into 0.5 mm quartz capillaries and measured for ≈ 20 min over the q-range 0.001 < q < 1 Å⁻¹. SAXS data were normalised using the SasView software[91] and scattering from the empty capillary and instrument was subtracted. The curves were fitted using Eq. (3)[10], which describes a porous matrix with random size and shape of pores based on the autocorrelation function derived by Debye et al.[7]:

$$I(q) = \frac{A}{\mathbf{q}^4} + \frac{B'a_1^4}{\left(1 + a_1^2\mathbf{q}^2\right)^2} + D \tag{3}$$

where $I$ is the intensity at a given $\mathbf{q}$ (scattering vector, $\mathbf{q} = 4\pi(\sin\theta/\lambda)$), being $\lambda$ the Cu Kα wavelength (1.5418 Å) and 2θ the scattering angle.

The first term, which describes the behaviour at low **q**, is the Porod scattering arising from the particle morphology. A is a scale factor related to the particle surface scattering (open porosity), B' is a scale factor related to pore scattering and proportional to the surface area of the closed pores; D is a constant background term and $a_1$ is a size factor related to the radius of a spherical pore.

The average diameter (D') of closed nanopores can be obtained from $a_1$ assuming spherical pores using the following equation[10]:

$$D' = 2 \times a_1 \times \sqrt{10} \qquad (4)$$

Furthermore, from B' it is possible to calculate the B'' value, which is proportional to the total number of closed pores[32]:

$$B'' = \frac{B'}{a_1^2} \qquad (5)$$

Gas sorption isotherms were measured on a Micromeritics 3 Flex 3500 gas sorption analyser using high-purity nitrogen gas at 77 K. Before the analysis, the materials were outgassed under a primary vacuum at 100 °C for 24 h.

## Electrochemical characterisation

The electrochemical performance of HC1000 and HC7000 pristine and ball-milled samples was tested in stainless-steel 2030-type coin cells (TOB New Energy) assembled in an argon-filled glovebox (MBraun) with $O_2$ and $H_2O \leq 0.1$ ppm. Electrode slurries were prepared by mixing a homogenous powder mixture (500 mg) consisting of 95% HC active material and 5% carboxymethyl cellulose (CMC, Sigma) binder with 1.3 mL $H_2O$ using a Thinky mixer (Intertronics) at 2000 rpm for 15 min. No balls were added to the mixing vessel to avoid microstructural particle changes during this process. The homogeneous slurry was cast onto Al foil (16 μm, 99.45%, TOB New Energy) using a doctor blade (MTI Corporation) and then dried at 60 °C for 12 h. Once dried, 12 mm diameter electrodes were cut and pressed under ≈ 100 MPa to reduce porosity and improve their conductivity. The average loading of the electrode active material on the current collector is 2–2.5 mg cm$^{-2}$. Sodium (ingot, 99.8% metal basis, Alfa Aesar) was made into metal discs (15 mm diameter, ca. 1 mm thickness), to use as counter and reference electrodes, where the disc surface was polished before use. A Whatman GF/B glass microfiber was used as a separator (1 μm thickness). In total, ≈ 150 μl of 1 M NaPF$_6$ (99+ %, Alfa Aesar) in the anhydrous organic solution of ethylene carbonate/diethylene carbonate (EC:DEC 1:1 w/w%, battery grade, Gotion) was used as the electrolyte. The electrolyte solvent was dried under molecular sieves (4 Å, Sigma) for a week before use. Before cycling, coin cells were rested for 10 h to obtain a stable open circuit voltage (OCV).

Galvanostatic charge/discharge cycling tests were carried out in the voltage window 3–0.01 V vs. Na$^+$/Na at 5 mA g$^{-1}$ and 50 mA g$^{-1}$ for 100 cycles. Rate performance tests were carried out in the voltage window 3–0.01 V vs. Na$^+$/Na from 5 to 100 mA g$^{-1}$. CV experiments were carried out in the voltage window 3–0.01 V vs. Na$^+$/Na at a scan rate of 0.1 mV s$^{-1}$. Galvanostatic and CV data were collected in a battery cycler (CT-4008Tn, Neware) and potentiostat (VMP300, Biologic), respectively, at ambient temperature.

## EPR spectroscopy

Room temperature EPR spectra were collected using a continuous-wave (CW) Bruker EMX micro spectrometer with a microwave frequency close to 9.8 GHz, modulation amplitude of 0.5 G, microwave power of 2.206 mW, and a receiver gain of 30 dB. Average spectra were collected after 10 scan repetitions. All the samples were kept under vacuum before and during the measurement.

Temperature-dependence EPR studies in the temperature range 10 K to 100 K were performed using a Bruker EMX micro spectrometer with a microwave frequency close to 9.4 GHz, modulation amplitude of 0.5 G, microwave power of 2.206 mW, and a receiver gain of 10 dB under non-saturating conditions. Average spectra were collected after 10 scan repetitions. Temperature calibration was carried out using a temperature sensor probe (Cernox®, Lake Shore Cryotronics, Inc.) inserted into the sample position.

For EPR spectra analysis, the DI value (absolute spectra area value) of the EPR signal was used throughout the whole manuscript. EPR spectra that followed the Curie–Weiss law can be fitted with the following equation:

$$\chi = A + C/(T - T_0) \qquad (6)$$

where $\chi$ is the spin density, which can be calculated from the double integrated value (the absolute area value) of the first-derivative of the EPR signal, A is the Pauli contribution, C is the Curie–Weiss constant, and $T_0$ is the Curie–Weiss temperature.

For ex situ EPR measurements, the electrode slurries (made of 95% wt. HC and 5%wt. polyvinylidene difluoride (PVDF, Sigma) were dissolved in 1.3 mL N-methyl-2-pyrrolidone (NMP) (anhydrous, > 99.0%, Alfa Aesar) and coated onto the polytetrafluoroethylene (PTFE) separator (Celgard). PTFE is EPR silent and has good flexibility to be placed into the EPR tube (3 mm diameter). The choice for PVDF binder rather than water-soluble carboxymethyl cellulose (CMC) was made based on the hydrophobic nature of PTFE. Cast electrodes were cycled to different states of charge at 5 mA g$^{-1}$, extracted from the coin cell and washed with diethyl ethylene carbonate (DEC, battery grade, Gotion) and dried at room temperature in an Ar-filled glovebox (MBraun) with $O_2$ and $H_2O \leq 0.1$ ppm. After this, samples were sealed into EPR quartz tubes before transferring them outside the glovebox for data collection.

The EPR lineshape simulation of a single component was performed in MATLAB[1] using the 1st derivative of the Dysonian equation (Eq. (7))[92]:

$$y = y_0 + C \times \frac{2w \times (x_c - x)}{\left((x_c - x)^2 + w^2\right)^2} + C \times a \times \frac{w^2 - (x_c - x)^2}{\left((x_c - x)^2 + w^2\right)^2} \qquad (7)$$

where $y$ is the signal intensity, $y_0$ is the background, $C$ is the amplitude factor, $x$ is the magnetic field, $x_c$ is the centre magnetic field of the EPR signal, $w$ is the linewidth, $a$ is the ratio of the dispersion to absorption with a range from 0 to 1. If $a = 0$, the EPR signal is the symmetric Lorentzian lineshape, whereas if $a > 0$, a Dysonian lineshape is considered. The Dysonian lineshape, with its characteristic asymmetric factor A/B, corresponds to the ratio of intensities of the spectral positive peak to the negative one.

For multi-component spectra fitting, this can be realised by mathematical addition, where $y_{total}$ is the sum of individual $y$ components obtained from Eq. (7), i.e., $y_{total} = y_1 + y_2$.

## Data availability

Data that support the findings of this study are available within this article and its Supplementary Information. Source data are provided with this paper.

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

## Acknowledgements

N.T.-R. would like to acknowledge the financial support of Lancaster University, Imperial College London, and the Faraday Institution (FIRG018 Next Generation Na-ion batteries and EP/T005394/1 FutureCat projects). EPR experiments were performed at the EPSRC (Engineering and Physical Sciences Research Council) National Research Facility for EPR at the University of Manchester (NS/A000055/1). The authors would like to thank Dr. Peter Curran (Deregallera) for the provision of the pristine hard carbon samples to conduct this work.

## Author contributions

B.W. and N.T.-R. designed the experiments and interpreted the data. J.R.F. collected and analysed the XRD data. A.B. helped B.W. collect the EPR data and A.J.F. assisted in the interpretation of the EPR data. J.T. acquired the Raman data. The XPS data were collected by B.F.S and analysed by B.W. and J.R.F. S.B. performed the SEM experiments. E.R. acquired the SAXS data and helped in the interpretation of the resulting data. J.E. performed the BET experiments. K.H. and C.M.K. synthesised the HC700 and HC1000 materials. B.W., J.R.F. and N.T.-R. wrote and edited the manuscript. N.T.-R. supervised the project.

## Competing interests

The authors declare no competing interests.
