## [Peer Review File · Nature Communications]

REVIEWER COMMENTS

Reviewer #1 (Remarks to the Author):

The manuscript investigates the sodium charge storage mechanism of hard carbon anodes through electron paramagnetic resonance (EPR) spectroscopy as diagnostic tools. The authors tried to distinguish the nature of the spins of hard carbon with different carbonaceous structures through the EPR line-shape simulation and temperature-dependent transformation rules of spinning electron. This work is technically sound to establish relationships between specific structure modification and sodium storage mechanism. We recognize the importance of developing new types of characterization techniques with EPR to precisely distinguish the controversial mechanism study of sodium ions storage in hard carbon. However, the manuscript was not well-presented with many doubts and issues to be furtherly addressed. We do think the novelties of the present job are not obvious. Thereby, we regrettably feel that the manuscript could not be published at Nature Communications at this format.

Our detailed comments are stated as below:

- 1) The most noteworthy highlight of this manuscript rests with the intended distinction of the EPR signal for the 'slope' and 'plateau' capacity ascription of different microstructure in hard carbon. However, the author mainly discussed the characteristic results of pristine materials with different modifications, whereas few results are presented of the discharged hard carbon at different charge/discharge states. This still makes it confuse to reasonably understand the storage mode of sodium ions within hard carbon during exact discharging process. To better support the authors' opinion, in-situ EPR test or more ex-situ EPR results should be furtherly conducted to monitor the evolution of sodium ion storage within hard carbon.
- 2) The author provided mass raw data in the main text without organized summary and reasonable presentation, which made it hard to be understood and awkward to read. It would be better if the author can furtherly enhance the relationship of different experimental data with more concise logicity and comprehension.
- 3) As the fitting of EPR signal is not a general knowledge, the author should interpret the data fitting principle and its detailed fitting process with reasonable equations like what you did for the SAXS data processing or the references published in the journal of Materials Today, 2018, 21, 231-240.
- 4) In the sentence of line128-129 on page 4, the author claimed that differences exist in the XRD diffraction peaks for the sample of B1810 and B187, however the results presented in the Figure SI-3 showed inexplicit shifts of (002) peak between different samples. Therefore, this data is not that convincing. It would be better for the author to provide other experimental data like TEM to support their conclusion.

5) In figure 1 e and j, the author didn't provide the definition of the parameters of a1 and also didn't conduct any discussion for the two pictures in the text. From the results in the two pictures, negligible divergences lie in the two series samples for their average pore size. Similar conditions also occurred in the Figure 4 d, the authors didn't explain anything for the presented results.

6) Results presented in the Figure a and b are reduplicated with the results in Figure 2 and 3.

7) The language should be checked again since there are some typos.

i.e., in the line 150 on page 5, "in HC materials synthesised.71 (B187) to 2.01 from other biowaste precursors."; in the line 397 on page 16, "...towards Curie-Weiss behaviour It is therefore not..."; in page 5b, the amplification coefficient of the signal intensity for pristine B187 seems not "x50" as the original signal of B187 is much higher than B1810.

Reviewer #2 (Remarks to the Author):

EPR is a well-known powerful technique for the characterization of carbon materials. Particularly, the cation storage properties have been the topic of many studies connected with their applicability in different metal-ion batteries. In this context, the manuscript by Bin Wang et al. describes a systematic approach to gaining valuable information from EPR data. The analysis focuses on two HC samples obtained from biowaste and thermally treated at 1000 °C and 700 °C and then subjected to two different milling times and speeds. The microscopic, XRD, gas adsorption, Raman, and XPS characterizations reveal significant differences between samples that are nicely correlated with their EPR spectra at different temperatures and the electrochemical performance in Na half-cells. From the ex-situ EPR data of electrodes at different states of charge, the authors find a clear correlation between the EPR spectral components and the mechanism of sodium storage in two different regions of the cell discharge curve. These conclusions are sound and represent a significant advance in the possibilities of the EPR technique to predict the electrochemical performance of HC electrodes. The publication of the manuscript in its present form is strongly recommended.

Electron Paramagnetic Resonance Spectroscopy as a Fast Diagnostic Tool to Determine the Sodium Charge Storage Mechanism of Hard Carbon Anodes

Point-by-point responses to reviewers:

We are very grateful to the reviewers for their time and careful consideration of our manuscript. We appreciate their advice in helping us deliver the best possible version of the manuscript. Please find our point-by-point responses to their comments.

All changes made to the manuscript and supporting information are indicated in blue text in the responses below and highlighted text in the respective documents themselves.

Reviewer #1: The manuscript investigates the sodium charge storage mechanism of hard carbon anodes through electron paramagnetic resonance (EPR) spectroscopy as diagnostic tools. The authors tried to distinguish the nature of the spins of hard carbon with different carbonaceous structures through the EPR line-shape simulation and temperature-dependent transformation rules of spinning electron. This work is technically sound to establish relationships between specific structure modification and sodium storage mechanism. We recognize the importance of developing new types of characterization techniques with EPR to precisely distinguish the controversial mechanism study of sodium ions storage in hard carbon. However, the manuscript was not well-presented with many doubts and issues to be furtherly addressed. We do think the novelties of the present job are not obvious. Thereby, we regrettably feel that the manuscript could not be published at Nature Communications at this format.

1) The most noteworthy highlight of this manuscript rests with the intended distinction of the EPR signal for the 'slope' and 'plateau' capacity ascription of different microstructure in hard carbon. However, the author mainly discussed the characteristic results of pristine materials with different modifications, whereas few results are presented of the discharged hard carbon at different charge/discharge states. This still makes it confuse to reasonably understand the storage mode of sodium ions within hard carbon during exact discharging process. To better support the authors' opinion, in-situ EPR test or more ex-situ EPR results should be furtherly conducted to monitor the evolution of sodium ion storage within hard carbon.

We collected *ex situ* EPR data for all the samples shown in this study and included a discussion in the manuscript regarding the post-mortem analysis for all these. We also improved the previous ex-situ EPR discussion for the two pristine samples in the manuscript to aid understanding from the readership.

Electron Paramagnetic Resonance Spectroscopy as a Fast Diagnostic Tool to Determine the Sodium Charge Storage Mechanism of Hard Carbon Anodes

New ex-situ EPR spectra for the ball-milled samples can be found in (Figure SI-14) and a new Table has been added to the SI (Table S7), to provide information on g values, peak-to-peak linewidth and lineshape fitting information for all the signals obtained during the ex-situ experiments.

Changes to the manuscript

(Page 12, line 12)

EPR spectrum of pristine HC700 at 10 K showed an intense and sharp signal with a Lorentzian lineshape centred at $g = 2.0024$ and $\Delta H_{pp} = 9.5$ G (Figure 5a) that obeyed the Curie-Weiss law, as expected from localised spin centres (Figure 6a).³⁸ The spin g-factor is close to the g-factor for a free electron ($g_e = 2.0023$), which indicates that the sample has a large spin-orbit coupling constant due to the presence of heteroatoms such as oxygen and nitrogen.³⁸ EPR parameters of the signal are consistent with localised radicals, which can be attributed to a defective carbon structure, such as dangling bonds with terminating oxygen/nitrogen groups (as observed in the XPS data) in the edge of the graphene sheets and/or on the surface of open micropores and vacancies within the graphene sheets.^{75, 76}

Through post-mortem EPR studies at 10 K, we studied the evolution of this resonance during the discharge process at 0.5 and 0.02 V using a constant current of 5 mA g⁻¹. Post-mortem EPR spectra are shown in Figures 7a and c. The EPR signals observed during discharge to 0.5 V and 0.02 V were similar to the initially observed narrow signal fitted to a Lorentzian lineshape in the pristine material, with g values close to 2.003 and Curie-Weiss type behaviour, indicating the nature of localised spins. A progressive line narrowing occurred when decreasing the potential (from 9.48 G for pristine HC700 to 8 G at 0.5 V, and 6.4 G at 0.02 V), in agreement with previous reported works.³³ The changes in the linewidth of the HC700 at different potentials may be explained by alterations in the environment of the free radicals related to the differences in the binding energy of defects/functional groups with Na⁺ ions.^{68, 69} The chemical process reflected in the EPR signals is directly echoed in the electrochemical curves with the presence of a slope, which has been previously explained with the sodiation of the dangling bonds over a range of different energies.^{43, 44} The localised free radicals in pristine HC700 act as electron acceptors towards Na⁺, forming covalent C-Na bonds. As the potential decreases, more of these C-Na bonds form, leading to a continuous decrease or even disappearance of original localized radicals in disordered carbons, as previous works have reported.³⁶ The reminiscent EPR signal observed in the spectra after discharge to 0.02 V may be explained by residual bonds which have not reacted with Na. Overall, it was not possible to provide a quantitative measurement of the spins in this study due to difficulties in preserving the electrodes intact during sample

Electron Paramagnetic Resonance Spectroscopy as a Fast Diagnostic Tool to Determine the Sodium Charge Storage Mechanism of Hard Carbon Anodes

preparation. Furthermore, *ex situ* EPR studies on the two ball-milled HC700 samples after discharge to 0.02 V showed similar EPR signals (**Figure SI-14, Table S7**), further revealing an analogous reaction mechanism to HC700. We did not observe an EPR signal related to the presence of metallic or quasi-metallic Na at 0.02 V which might be related to the insertion of Na into the closed pores or Na plating, despite the observance of a small voltage plateau in the first discharge, which has been attributed to these processes (**Figure 3c**).^{17, 77} Previous *ex situ* EPR studies have explained the presence of metallic or quasi-metallic Na with the presence of sharp resonances in the spectra. Nevertheless, these signals were observed in HCs synthesised at higher temperatures than 700 °C and thus, with significantly different electrochemical signatures than HC700, including longer plateaus > 100 mAh g⁻¹.^{12, 78} Furthermore, data were collected at different electrochemical conditions to those reported in this work, including the over-discharge of the electrode or data collected during the second discharge cycle. Alternatively, it is possible that this signal is weak and overlaps with the narrow signal attributed to the free radicals and therefore it is not possible to overrule this process.

(Page 13, line 20)

Increasing the temperature from 700 °C to 1000 °C led to a drastic reduction of the EPR signal intensity by a factor of 8, which we attributed to the start of the graphitisation process and subsequent recombination of free radicals and removal of defects. A broad signal with an asymmetric (Dysonian) lineshape ($A/B = 1.07$) ($g = 2.0044$ and $\Delta H_{pp} \approx 150$ G) at 10 K (**Figure 5d**) and paramagnetic Pauli behaviour was observed (**Figure 6c**).⁷⁴ This signal is attributed to the existence of mobile electrons from intrinsic defects from basal planes in the carbon structure, on which electrons can move without limitation.^{73, 74} Therefore, it is plausible to relate this signal to storage processes where Na⁺ ions might interact directly or indirectly with these basal planes (i.e., through Na⁺ intercalation between graphitic planes or pore-filling insertion^{17, 66}) and thus, we might associate this signal to the plateau observed in the galvanostatic curves for this sample. It is worth noting that the broadening of the EPR signal did not arise from the presence of O₂ molecules adsorbing to the open pores of the HC sample, as seen in other works,^{36, 41} since data were collected under vacuum.

Through post-mortem EPR analysis we also studied the evolution of this resonance at different states of charge. Upon discharge to 0.5 V, the initial EPR signal of HC1000 with Pauli behaviour disappeared, and two new signals with g values close to 2.004 with different linewidths ($\Delta H_{pp} = 3$ G and 57 G) and displaying Curie-Weiss behaviour appeared (**Figure 7 b and d**), similar to previously reported results.^{43, 78} The broader signal decreased in linewidth to 32 G and narrow/broad signal ratios increased from 0.28 to 0.38 when the

Electron Paramagnetic Resonance Spectroscopy as a Fast Diagnostic Tool to Determine the Sodium Charge Storage Mechanism of Hard Carbon Anodes

voltage decreased from 0.5 to 0.02 V. Alkaline ion insertion into graphitic layers has been thought to show an increased EPR signal during the discharging process due to the spin-orbit interactions between intercalated Na/Li and the delocalized π -electrons graphite crystallites.^{43, 79, 80} Thus, an explanation for these observations is that the broad component in the discharged HC1000 sample arises from localised spins when Na⁺ ions are intercalated into the graphitic structure, and the narrow signal is associated to the formation of localized paramagnetic centres after Na adsorbed on the disordered carbon surface.⁴⁰ This is further supported by the increased broad/narrow ratio and its decreased linewidth feature over the discharging process, as the increased spin concentration at lower voltage causes further linewidth narrowing. Akin to the HC700 ex-situ studies, we could not conclusively determine whether Na quasi-metallic related to Na insertion into the closed pores was present in this sample.

(Page 15, line 3)

..... For the ball-milled HC1000 samples, the linewidths of the broader signal exhibited an opposite trend to the HC700 samples, as the linewidth decreased from 150 G (HC1000) to 140 G (HC1000-400-2h) and 25 G (HC1000-400-5h) (**Figure 5 d-f** and **Table S6**). Ball-milling causes a reduction in the range of the extended aromatic structure, which is accompanied by a change in the spin nature in these samples, i.e., from Pauli to Curie-Weiss law behaviour (**Figure 6c**). This behaviour might be related to carbon structure degradation, which has proved to be detrimental to the electrochemical performance observed in the HC1000-400-5h sample. Ex-situ EPR studies also reflected this unusual behaviour, where only one signal was observed after discharge to 0.02 V, suggesting one Na storage process related to the slope capacity behaviour. This contrasts with the two signals observed in HC100-400-2h (**Figure SI-14**) and HC1000 (**Figure 7**), which featured two different Na storage behaviours which might be attributed to both slope and plateau processes.

For the ball-milled HC1000 samples, the linewidths of the broader signal exhibited an opposite trend to the HC700 samples, as the linewidth decreased from 150 G (HC1000) to 140 G (HC1000-400-2h) and 25 G (HC1000-400-5h) (**Figure 5 d-f** and **Table S6**). Ball-milling causes a reduction in the range of the extended aromatic structure, which is accompanied by a change in the spin nature in these samples, i.e., from Pauli to Curie-Weiss law behaviour (**Figure 6c**). This behaviour might be related to carbon structure degradation, which has proved to be detrimental to the electrochemical performance observed in the HC1000-400-5h sample. Ex-situ EPR studies also reflected this unusual behaviour, where only one signal was observed after discharge to 0.02 V, suggesting one Na storage process related to the slope capacity behaviour. This contrasts with the two signals observed in

Electron Paramagnetic Resonance Spectroscopy as a Fast Diagnostic Tool to Determine the Sodium Charge Storage Mechanism of Hard Carbon Anodes

HC100-400-2h (Figure SI-14) and HC1000 (Figure 7), which featured two different Na storage behaviours which might be attributed to both slope and plateau processes.

Changes to the SI

Figure SI-14. *ex situ* EPR measurements of ball-milled (a) HC700 and (b) HC1000 samples when discharged to 0.02 V.

Electron Paramagnetic Resonance Spectroscopy as a Fast Diagnostic Tool to Determine the Sodium Charge Storage Mechanism of Hard Carbon Anodes

Table S7. *g* value, linewidth and lineshape information obtained from the ex-situ EPR spectra measured at 10K.

Sample@ 10K	Broad signal			Narrow signal		
	g	$\Delta H_{pp}/G$	Lineshape	g	$\Delta H_{pp}/G$	Lineshape
HC700-400-2h @ 0.02 V	-	-		2.0032	6.3	L*
HC700-400-5h @ 0.02 V	-	-		2.0029	9.6	L*
HC1000-400-2h @ 0.02 V	2.0041	12.5	L*	2.0035	5.2	L*
HC1000-400-5h @ 0.02 V	-	-		2.0037	7.5	L*

* L =Lorentzian lineshape

2) The author provided mass raw data in the main text without organized summary and reasonable presentation, which made it hard to be understood and awkward to read. It would be better if the author can furtherly enhance the relationship of different experimental data with more concise logicity and comprehension.

To aid the organisation, comprehension, and presentation of our results we conducted an extensive re-writing of the manuscript. Due to this extensive re-writing, we have not tracked changes in the manuscript related to changes in the writing style and have highlighted in yellow added information or changes that we have added to the manuscript to draw the attention of the reviewer. In particular, we have:

Changes to the manuscript:

- Renamed our samples from B187 and B1810 to HC700 and HC1000, respectively, so the pyrolysis temperature at which these samples were produced can be easily identified by the reader when going through the manuscript.
- Removed two datasets of the manuscript (those related to the B187, and B1810 HCs being milled at 600 rpm for 2h) to focus only on the influence of the ball-milling time (and not the speed) on structure, electrochemistry and EPR data of the HC materials studied. We

Electron Paramagnetic Resonance Spectroscopy as a Fast Diagnostic Tool to Determine the Sodium Charge Storage Mechanism of Hard Carbon Anodes

considered that having these two datasets was making the manuscript longer without adding any value to the general discussion of results, as data were almost identical to that observed for B187 and B1810 at 400 rpm for 2h.

- We have restructured the article by adding several subheadings in the “Results and Discussion” section (see manuscript) and swapped the order of the EPR and electrochemistry section so we could easily explain, and correlate data related to XRD, XPS, Raman, SAXS and BET to the electrochemical behaviour of all the studied materials first, and then we could further relate these with the EPR data, which is the focus of our manuscript. We thought it could have been confusing for the reviewer to find the EPR data first before understanding how the samples perform electrochemically.

- We have updated or incorporated new summary tables and figures in the manuscript that compare and/or combine results from different techniques to aid understanding and have incorporated new discussion to reflect the new data shown in the Figures/Tables. These are:

- **Table 1-** which summarises relevant parameters from XRD, XPS, Raman, SAXS and BET.

Table 1. Summary of selected structural parameters of pristine and ball-milled HC700 and HC1000 samples obtained from XRD, XPS, Raman, SAXS and BET data.

Sample	XRD	XPS				Raman	SAXS				BET
	d_{002}/nm	C/%	O/%	N/%	C/O	I_{D3}/I_G	$a_1/\text{a.u.}$	$A/\text{a.u.}$	$B'/\text{a.u.}$	$B''/\text{a.u.}$	$S/\text{m}^2 \text{g}^{-1}$
HC700	0.38	92.8	3.3	3.9	25.6	1.01	3.19	3.2E-6	1.0E-4	9.9E-6	326.9
HC700-400-2h	0.37	91.1	4.9	3.9	16.2	1.05	3.56	4.1E-6	8.8E-5	6.9E-7	211.6
HC700-400-5h	0.37	87.7	8.7	3.6	11.4	1.24	4.24	1.7E-5	1.2E-5	6.7E-8	107.2
HC1000	0.37	97.1	1.5	1.4	57	0.65	3.95	8.0E-6	2.1E-4	1.4E-5	27.1
HC1000-400-2h	0.37	94.4	4.4	1.2	19.3	0.67	4.08	7.1E-6	1.1E-4	6.5E-6	22.4
HC1000-400-5h	0.37	88.2	10	1.8	9	0.76	5.39	1.2E-5	9.3E-5	3.2E-6	203.9

(Page 4, line 15)- Text discussing d_{002} trend among all the studied samples

The average interlayer distance (d_{002}) in both pristine HC700 and HC1000 materials was calculated as 0.38 nm and 0.37 nm, respectively (**Table 1**). These values are larger than those of graphite (0.33 nm).⁴⁸ The difference in d_{002} values between both samples is expected to be small given the carbonisation temperatures used, as previous findings have shown that the d_{002} values of HCs only begin to significantly decrease above synthesis temperatures of 1400 °C.²⁸ Ball-milling these samples resulted only in almost negligible changes in the d_{002} interlayer spacing (**Table 1**) and subtle changes in crystallinity with

Electron Paramagnetic Resonance Spectroscopy as a Fast Diagnostic Tool to Determine the Sodium Charge Storage Mechanism of Hard Carbon Anodes

milling conditions (as reflected by the FWHM calculated values for these samples, **Table S1-1**), as shown in earlier reports.³²

(Page 4, line 33)- Text discussing I_{D3}/I_G trend among all the studied samples

...while the I_{D3}/I_G peak area ratio provided an indication of the oxygen-containing functional groups in the samples.⁵¹ I_{D1}/I_G and I_{D3}/I_G values are shown in **Figure 2a** and **Table 1**. The higher carbonisation temperature used to produce HC1000 led to lower I_{D1}/I_G and I_{D3}/I_G ratios, showing a higher degree of in-plane ordering and a lesser presence of oxygen-containing terminal groups, respectively. This behaviour was analogous to that observed in other HC materials synthesised from biowaste precursors.^{17, 48} Upon extending the milling time, I_{D1}/I_G and I_{D3}/I_G ratios increased in both cases due to higher defect site concentration and the presence of carbon-oxygen linkages.²⁶ Similarly, L_a values calculated using **Equation S2** indicated a reduction of the graphitic domains upon milling, as expected from particle fragmentation caused by this treatment (**Figure 2b**).

(Page 7, line 13)- Text discussing B'' trend among all the studied samples

The destruction of some of the closed porosity with the mechanical treatment is responsible for the observed reduction in the surface area of the closed pores (B' in **Equation S3**) and the relative number of pores (B'' in **Equation S5**) (**Table 1**).

- **Figure 3c and f** and **Table S3**- which summarise slope, plateau and total capacities in the 1st and 2nd charge cycles and ICE.

Electron Paramagnetic Resonance Spectroscopy as a Fast Diagnostic Tool to Determine the Sodium Charge Storage Mechanism of Hard Carbon Anodes

Figure 3. Slope and plateau capacity contribution to the total capacity during 1st charge and 2nd charge processes for (e) HC700 samples and (f) HC1000 samples.

Table S3. Slope and plateau capacity values obtained from the 1st and 2nd charge processes of the pristine and ball-milled HC700 and HC1000 samples using dQ/dV vs. V curves shown in Figure SI-10.

Sample	Charge Cycle	Capacity/mAh g ⁻¹			Onset Voltage/mV	ICE/%
		Slope	Plateau	Total		
HC700	1 st	114	23	137	105.9	53.6
	2 nd	108	25	133	106.3	89.2
HC700-400-2h	1 st	151	27	178	114.2	52.3
	2 nd	146	27	173	114.0	87.9
HC700-400-5h	1 st	227	24	251	109.1	50.5
	2 nd	218	24	242	112.3	88.9
HC1000	1 st	142	87	229	94.2	67.2
	2 nd	139	87	226	93.3	93.4
HC1000-400-2h	1 st	135	80	215	94.5	65.1
	2 nd	130	83	213	95.1	93.2
HC1000-400-5h	1 st	246	8	254	70.2	51.7
	2 nd	232	8	240	72.1	88.5

Electron Paramagnetic Resonance Spectroscopy as a Fast Diagnostic Tool to Determine the Sodium Charge Storage Mechanism of Hard Carbon Anodes

- **Figures 2a, b and c-** which show trends in I_{D1}/I_G , L_a and D' parameters obtained from Raman and SAXS data.

Figure 2. (a) I_{D1}/I_G and (b) L_a values calculated from Raman spectroscopy data; and (c) D' values calculated from SAXS data for pristine and ball-milled HC700 and HC1000 samples.

Text discussing “ L_a ” and D' in Figure 2 (I_{D1}/I_G value had been discussed previously):

(Page 5, line 7)

.....Similarly, L_a values calculated using **Equation S2** indicated a reduction of the graphitic domains upon milling, as expected from particle fragmentation caused by this treatment (**Figure 2b**).

(Page 7, line 7)

The diameter of the closed pores (D') (which is proportional to a_1 which is a size factor related to the radius of a spherical pore, as shown in **Equation S4**), increased with increasing temperature, where pore dimensions increased from 20 to 27 Å in HC700 and 25 to 31 Å in HC1000, due to the growth and alignment of the graphitic domains (**Figure 2c**).

- We added a diagram in **Figure 2d** to better illustrate the effects of milling in the open and closed porosity.

Electron Paramagnetic Resonance Spectroscopy as a Fast Diagnostic Tool to Determine the Sodium Charge Storage Mechanism of Hard Carbon Anodes

Figure 2(d). Schematic showing the effects on open and closed porosity in HC samples after the ball-milling treatment.

- We added new text in the “Electrochemical performance of pristine and ball-milled HC700 and HC1000 samples” section to summarise and correlate the structural data shown in the “Structural characterisation of pristine and ball-milled HC700 and HC1000 samples” section

(Page 9, line 25):

Combining our previous structural and electrochemical analysis we can establish that ball-milling destroys the amount/size of the ordered graphitic regions within the HC structure, as evidenced by the decrease in crystallinity (XRD) and the increase in defect concentration (Raman spectroscopy). This resulted in a decreased particle size (SEM) and a subsequent increase in open porosity (BET surface area and A parameter from SAXS), oxygen content at the surface of the HC particles (XPS, Raman spectroscopy), and diameter of the close pores (D' parameter in SAXS), attributed to pore opening, followed by pore coalescence and subsequent formation of new pores. These structural changes were expressed electrochemically by a decrease in ICE, due to increased SEI formation, and increased dominance of the sloping capacity, attributed to increased Na⁺ adsorption at the surface due to the presence of functional groups and defects. The general decrease in capacity retention observed with ball-milling time suggested that Na⁺ adsorption at these surface sites, which is promoted by ball-milling, is not as reversible over a long-time scale compared to the storage processes occurring in the plateau region, supporting previous literature reports.⁷⁰

Electron Paramagnetic Resonance Spectroscopy as a Fast Diagnostic Tool to Determine the Sodium Charge Storage Mechanism of Hard Carbon Anodes

- We simplified the EPR results and discussion section in the manuscript by discussing only the spectra obtained at 10 K, and not at room temperature, since the same trends are observed in both sets and only different linewidth values are observed in the signals due to the non-freezing of the spins, unlike at 10 K. Therefore, discussing data at room temperature does not add any value or provides any further insights to the discussion. Nevertheless, we decided to keep the EPR data related to room temperature data from the previous manuscript version in the SI (**Figure SI-13 and Table S5**). Furthermore, we included a new Table (**Table S6**) which summarises g values, linewidth and lineshape information for the EPR signals obtained at 10 K to aid understanding of the EPR data presented in the manuscript.

Table S6. g value, linewidth and lineshape information obtained from the EPR spectra measured at 10K.

Sample@ 10K	Broad signal			Narrow signal		
	g	$\Delta H_{pp}/G$	Lineshape	g	$\Delta H_{pp}/G$	Lineshape
HC700	2.0024	9.5	L*	-	-	-
HC700-400-2h	2.0026	26.5	L*	2.0036	4.9	L*
HC700-400-5h	2.0025	72.9	L*	2.0026	9.12	L*
HC1000	2.0044	150	D (1.07) **	-	-	-
HC1000-400-2h	2.0025	140	D (1.03) **	2.003	6.1	L*
HC1000-400-5h	2.0027	24.9	D (1.04) **	2.0034	6.9	L*

* L =Lorentzian lineshape

** D= Dysonian lineshape, the value in brackets indicates the asymmetry parameter A/B, where A and B are the amplitudes of the positive and negative parts of the signal.

3) As the fitting of EPR signal is not a general knowledge, the author should interpret the data fitting principle and its detailed fitting process with reasonable equations like what you did for the SAXS data processing or the references published in the journal of Materials Today, 2018, 21, 231-240.

Changes to the SI:

We have included in the *Electron Paramagnetic Resonance (EPR) Spectroscopy section of the SI* information re. the fitting of the EPR spectra shown in the manuscript. The fitting shown in the Materials Today paper cited by the Reviewer is much more complex than the fitting used in this paper due to the presence of the Li signal, which causes a phase shift in

Electron Paramagnetic Resonance Spectroscopy as a Fast Diagnostic Tool to Determine the Sodium Charge Storage Mechanism of Hard Carbon Anodes

the overall signal. Consequently, it becomes necessary to isolate the carbon signal within the lithium and carbon mixture. The challenging aspect lies in the fact that the lineshape or asymmetric factor differs in each signal, necessitating calibration and a simulation process, as outlined in the supplementary information in the Materials Today paper.

In contrast, our conditions are simpler (as described below) due to the absence of the Li signal. A simulation involving two components is deemed sufficient for our purposes.

Page 5, Methods Section

The EPR lineshape simulation of a single component was performed in MATLAB¹ using the following equation:⁹

$$y = y_0 + C \times \frac{2w \times (x_c - x)}{((x_c - x)^2 + w^2)^2} + C \times a \times \frac{w^2 - (x_c - x)^2}{((x_c - x)^2 + w^2)^2} \quad (\text{Equation S7})$$

where y is the signal intensity, y_0 is the background, C is the amplitude factor, x is the magnetic field, x_c is the centre magnetic field of the EPR signal, w is the linewidth, a is the ratio of the dispersion to absorption with a range from 0 to 1. If $a = 1$, the EPR signal is the symmetric Lorentzian lineshape. If $a \neq 1$, a Dysonian lineshape is considered. The Dysonian lineshape, with its characteristic asymmetric factor A/B , corresponds to the ratio of intensities of the spectral positive peak to the negative one.

For multi-component spectra fitting, this can be realized by mathematical addition, where y_{total} is the sum of individual y components obtained from Equation S7, i.e., $y_{total} = y_1 + y_2$

We also included a new column, labelled “Lineshape” in Tables S5-S7 to summarise the lineshape fitting used for the different EPR signals observed in the spectra of all the samples.

Table S5. g value, linewidth and lineshape information obtained from the EPR spectra measured at room temperature and shown in Figure S13.

Sample@ RT	Broad signal			Narrow signal		
	g	$\Delta H_{pp}/G$	Lineshape	g	$\Delta H_{pp}/G$	Lineshape
HC700	2.0025	32	L*	-	-	
HC700-400-2h	2.0021	54	L*	2.0023	5.8	L*
HC700-400-5h	2.0024	118	L*	2.0031	10.3	L*
HC1000	2.0033	210	D (1.12) **	-	-	-
HC1000-400-2h	2.0029	200	D (1.01) **	2.0026	6.5	L*

Electron Paramagnetic Resonance Spectroscopy as a Fast Diagnostic Tool to Determine the Sodium Charge Storage Mechanism of Hard Carbon Anodes

HC1000-400-5h	2.0029	27.7	D (1.1) **	2.0029	6.4	L*
---------------	--------	------	------------	--------	-----	----

* L =Lorentzian lineshape

** D= Dysonian lineshape, the value in brackets indicates the asymmetry parameter A/B, where A and B are the amplitudes of the positive and negative parts of the signal.

Table S6. *g value, linewidth and lineshape information obtained from the EPR spectra measured at 10K shown in Figure 5.*

Sample@ 10K	Broad signal			Narrow signal		
	g	$\Delta H_{pp}/G$	Lineshape	g	$\Delta H_{pp}/G$	Lineshape
HC700	2.0024	9.5	L*	-	-	-
HC700-400-2h	2.0026	26.5	L*	2.0036	4.9	L*
HC700-400-5h	2.0025	72.9	L*	2.0026	9.12	L*
HC1000	2.0044	150	D (1.07) **	-	-	-
HC1000-400-2h	2.0025	140	D (1.03) **	2.003	6.1	L*
HC1000-400-5h	2.0027	24.9	D (1.04) **	2.0034	6.9	L*

* L =Lorentzian lineshape

** D= Dysonian lineshape, the value in brackets indicates the asymmetry parameter A/B, where A and B are the amplitudes of the positive and negative parts of the signal.

Table S7. *g value, linewidth and lineshape information obtained from the ex situ EPR spectra measured at 10K.*

Sample@ 10K	Broad signal			Narrow signal		
	g	$\Delta H_{pp}/G$	Lineshape	g	$\Delta H_{pp}/G$	Lineshape
HC700-400-2h @ 0.02 V	-	-	-	2.0032	6.3	L*
HC700-400-5h @ 0.02 V	-	-	-	2.0029	9.6	L*
HC1000-400-2h @ 0.02 V	2.0041	12.5	L*	2.0035	5.2	L*
HC1000-400-5h @ 0.02 V	-	-	-	2.0037	7.5	L*

* L =Lorentzian lineshape

4) In the sentence of line 128-129 on page 4, the author claimed that differences exist in the XRD diffraction peaks for the sample of B1810 and B187, however the results presented in the Figure SI-3 showed inexplicit shifts of (002) peak between different samples.

Electron Paramagnetic Resonance Spectroscopy as a Fast Diagnostic Tool to Determine the Sodium Charge Storage Mechanism of Hard Carbon Anodes

Therefore, this data is not that convincing. It would be better for the author to provide other experimental data like TEM to support their conclusion.

We thank the reviewer for this comment, which allowed us to reflect and improve the discussion regarding the XRD data obtained for our samples in more detail. For reference, the previous text in the initial submission read as follows:

Powder X-ray diffraction (PXRD) data of pristine and ball-milled B1810 and B187 samples showed two main broad reflections at 24° and 43.2° 2θ , characteristic for (002) and (100) Bragg peaks of graphite (Figure SI-3).¹⁷ The (002) diffraction peak in pristine B1810 occurs at a higher 2θ angle than in B187, leading to decreased averaged d_{002} interlayer spacing of the sp^2 carbon layers, as expected from the higher pyrolysis temperature used.⁴⁴ According to Bragg's law ($2d\sin\theta = n\lambda$), the interlayer distances (d_{002}) in both pristine B1810 and B187 samples were ca. 3.72 and 3.81 Å, respectively. These d_{002} values are much larger than that of graphite (3.33 Å),⁴⁵ thus, enabling Na^+ ion intercalation. No significant changes in the d_{002} interlayer spacing were found in both samples after the ball-milling treatments used in this work. We observed, however, subtle changes in crystallinity with milling conditions, as shown in earlier reports.²⁷ For example, B1810 showed decreased crystallinity with increasing milling time (B1810-400-5h), as reflected by the increased broadening of the diffraction peaks.

Figure SI-3. Powder XRD data of (a) B1810 and (b) B187 HC ball-milled samples.

In the revised version of the manuscript, we have included new XRD data which was obtained by re-measuring all the samples in a diffractometer for longer times (to minimise

Electron Paramagnetic Resonance Spectroscopy as a Fast Diagnostic Tool to Determine the Sodium Charge Storage Mechanism of Hard Carbon Anodes

noise). We also used Ti foil as a reference in our diffractograms to obtain accurate values of 2θ degrees for the 002 reflections in all the samples. New XRD data can be found in **Figure SI-3** and d_{002} values are included in **Table 1**. The new measurement protocol used for the measurement and calculation of the d_{002} interlayer distance is described in the Methods section of the SI (page 2). **Figure SI-4** was included in the SI to show an example of one of the fittings made to calculate d_{002} . Based on the d_{002} values obtained, now calculated more confidently by us, we concluded that there were minimal differences between all the samples studied in this work. This supports literature findings on d_{002} values for hard carbon material synthesised at the temperatures described here, which showed very minor differences in d_{002} values as well as XRD data on ball-milled samples previously reported in the literature. Thus, we added the text below in the manuscript to describe these observations.

Changes to the manuscript (Page 7):

Table 1. Summary of selected structural parameters of pristine and ball-milled HC700 and HC1000 samples obtained from XRD, XPS, Raman, SAXS and BET data.

Sample	XRD d_{002}/nm	XPS				Raman I_{D3}/I_G	SAXS				BET $S/\text{m}^2 \text{g}^{-1}$
		C/%	O/%	N/%	C/O		$a_1/\text{a.u.}$	$A/\text{a.u.}$	$B'/\text{a.u.}$	$B''/\text{a.u.}$	
HC700	0.38	92.8	3.3	3.9	28.1	1.01	3.19	3.2E-6	1.0E-4	9.9E-6	326.9
HC700-400-2h	0.37	91.1	4.9	3.9	18.6	1.05	3.56	4.1E-6	8.8E-5	6.9E-7	211.6
HC700-400-5h	0.37	87.7	8.7	3.6	10.1	1.24	4.24	1.7E-5	1.2E-5	6.7E-8	107.2
HC1000	0.37	97.1	1.5	1.4	64.7	0.65	3.95	8.0E-6	2.1E-4	1.4E-5	27.1
HC1000-400-2h	0.37	94.4	4.4	1.2	21.4	0.67	4.08	7.1E-6	1.1E-4	6.5E-6	22.4
HC1000-400-5h	0.37	88.2	10	1.8	8.7	0.76	5.39	1.2E-5	9.3E-5	3.2E-6	203.9

Changes to the manuscript (Page 4, line 13):

*Powder X-ray diffraction (PXRD) data of pristine and ball-milled HC700 and HC1000 samples showed two main broad reflections at ca. 24° and 44° 2θ , which are characteristic of (002) and (100) Bragg peaks of graphite (**Figure SI-3**).¹² The average interlayer distance (d_{002}) in both pristine HC700 and HC1000 materials was calculated as 0.38 nm and 0.37 nm, respectively (**Table 1**). These values are larger than those of graphite (0.33 nm).⁴⁸ The difference in d_{002} values between both samples is expected to be small given the carbonisation temperatures used, as previous findings have shown that the d_{002} values of HCs only begin to significantly decrease above synthesis temperatures of 1400°C .²⁸ Ball-milling these samples resulted only in almost negligible changes in the d_{002} interlayer spacing (**Table 1**) and subtle changes in crystallinity with milling conditions (as reflected by the FWHM calculated values for these samples, **Table SI-1**), as shown in earlier reports.³²*

Electron Paramagnetic Resonance Spectroscopy as a Fast Diagnostic Tool to Determine the Sodium Charge Storage Mechanism of Hard Carbon Anodes

Changes to the SI:

(Page 2, Methods section)

“Powder X-ray diffraction (XRD) data were collected on a Rigaku MiniFlex600 at room temperature in Bragg-Brentano geometry with a 0.6 kW Cu-source generator ($K_{\alpha} = 1.54059 \text{ \AA}$) and a D/teX Ultra detector. All samples were packed on a flat circular plastic insert placed inside a stainless-steel sample holder. To account for differences in sample packing/height, titanium foil was used as a reference for all of the samples. A circular disc of Ti foil was first measured on its own and used as a reference. Subsequent HC powdered samples were packed on top of the Ti foil. Thus, Ti reflections were present in all the X-ray diffraction data. All the XRD data were shifted so that the most intense Ti reflection was at $38.21^{\circ} 2\theta$ to match that of the pristine Ti foil reference sample. All samples were scanned from 5° to $70^{\circ} 2\theta$. All samples were scanned from 5° to $70^{\circ} 2\theta$. The average interlayer spacing was calculated according to Bragg’s law:

$$2d_{002} \sin \theta = n\lambda \quad (\text{Equation S1})$$

where d_{002} is the average interlayer spacing, θ is the diffraction angle of the (002) reflection, λ is the wavelength of the X-ray beam (Cu K_{α} radiation, 0.154059 nm) and n is the order of the reflection. To find accurately the position of the (002) reflection, a Gaussian peak was fitted to the (002) reflection and an asymmetric least squares smoothing baseline was used using the Origin software.¹ An example of such a fitting can be observed in **Figure SI-4**.

(Page 8 in SI)

Electron Paramagnetic Resonance Spectroscopy as a Fast Diagnostic Tool to Determine the Sodium Charge Storage Mechanism of Hard Carbon Anodes

Figure SI-3. Powder XRD data of pristine and ball-milled (a) HC700 and (b) HC1000 samples. Peaks assigned to the Ti foil have been labelled with a (*) symbol and peaks assigned to zirconium oxide (ZrO_2) are labelled with a (Δ) symbol. The ZrO_2 present in the ball-milled samples is due to the use of ZrO_2 balls for the ball-milling procedure. Ti reflections are present as Ti foil was used as reference material to obtain accurate 2θ values for the different samples.

Figure SI-4. Example of a fit of the (002) reflection for pristine HC1000 to ascertain the position, FWHM and d_{002} values.

(Page 17 in SI)

Table S1. Positions of the (002) reflection (in 2θ degrees) and FWHM values of HC700 and HC1000 pristine and ball-milled samples obtained from XRD data shown in **Figure SI-3**.

Sample	(002) $^{\circ}$ 2θ	FWHM/ 2θ
HC700	23.7 ± 0.18	7.5 ± 0.4
HC700-400-2h	23.9 ± 0.11	7.6 ± 0.3
HC700-400-5h	24.06 ± 0.011	7.93 ± 0.03
HC1000	24.2 ± 0.13	6.9 ± 0.3
HC1000-400-2h	24.1 ± 0.17	7.2 ± 0.4
HC1000-400-5h	23.872 ± 0.004	7.98 ± 0.011

5) In figure 1 e and j, the author didn't provide the definition of the parameters of a1 and also didn't conduct any discussion for the two pictures in the text.

Electron Paramagnetic Resonance Spectroscopy as a Fast Diagnostic Tool to Determine the Sodium Charge Storage Mechanism of Hard Carbon Anodes

The definition of the “ a_1 ” parameter was provided in the Supplementary information (Page 3). As reflected in **Equation S4** of the SI (Equation S2 in the first submission), D' (diameter of the closed pores) is proportional to a_1 , and therefore, we find that the D' parameter is more relevant to discuss in the manuscript than the a_1 coefficient. Therefore, we prepared a new figure (**Figure 2c- shown above**) to show trends in this parameter for all the samples. Nevertheless, we have added in the manuscript a sentence to clarify that a_1 and D' are proportional to each other. We also added the definition of a_1 in the manuscript.

Changes to the manuscript (Page 7, line 7):

*The diameter of the closed pores (D') (which is proportional to a_1 (size factor related to the radius of a spherical pore, as shown in **Equation S4**),*

From the results in the two pictures, negligible divergences lie in the two series samples for their average pore size.

To better highlight the differences in a_1 values for the different datasets we have decided to provide these values in Table 1 and remove Figures 1e and j to avoid any confusion by the readership.

Changes to the manuscript (Page 7):

Table 1. Summary of selected structural parameters of pristine and ball-milled HC700 and HC1000 samples obtained from XRD, XPS, Raman, SAXS and BET data.

Sample	XRD	XPS				Raman	SAXS				BET
	d_{002}/nm	C/%	O/%	N/%	C/O	I_{D3}/I_G	$a_1/a.u.$	A/a.u.	B'/a.u.	B''/a.u.	$S/m^2 g^{-1}$
HC700	0.38	92.8	3.3	3.9	25.6	1.01	3.19	3.2E-6	1.0E-4	9.9E-6	326.9
HC700-400-2h	0.37	91.1	4.9	3.9	16.2	1.05	3.56	4.1E-6	8.8E-5	6.9E-7	211.6
HC700-400-5h	0.37	87.7	8.7	3.6	11.4	1.24	4.24	1.7E-5	1.2E-5	6.7E-8	107.2
HC1000	0.37	97.1	1.5	1.4	57	0.65	3.95	8.0E-6	2.1E-4	1.4E-5	27.1
HC1000-400-2h	0.37	94.4	4.4	1.2	19.3	0.67	4.08	7.1E-6	1.1E-4	6.5E-6	22.4
HC1000-400-5h	0.37	88.2	10	1.8	9	0.76	5.39	1.2E-5	9.3E-5	3.2E-6	203.9

Similar conditions also occurred in the Figure 4 d, the authors didn't explain anything for the presented results.

As suggested by Reviewer 1, we have added more discussion in the manuscript related to Figure 4d (**Figure 7b** in the current submission). Figure 7b shows the temperature dependence behaviour of the simulated EPR data for the un-milled HC700 sample when this is discharged to 0.02 V. This Figure is shown to demonstrate that the susceptibility of the sample obeys the Curie-Weiss law (**Equation S6**) as reflected by the good fit between raw

Electron Paramagnetic Resonance Spectroscopy as a Fast Diagnostic Tool to Determine the Sodium Charge Storage Mechanism of Hard Carbon Anodes

and fitted data. We also added some discussion in the manuscript to explain what “obeying the Curie-Weiss law” as well as “Pauli behaviour” mean for a general non-specialist readership.

Changes to the manuscript (Page 12, line 21):

*Through post-mortem EPR studies at 10 K, we studied the evolution of this resonance during the discharge process at 0.5 and 0.02 V using a constant current of 5 mA g⁻¹. Post-mortem EPR spectra are shown in **Figures 7a and c**. The EPR signals observed during discharge to 0.5 V and 0.02 V were similar to the initially observed narrow signal fitted to a Lorentzian lineshape in the pristine material, with g values close to 2.003 and Curie-Weiss type behaviour, indicating the nature of localised spins. A progressive line narrowing occurred when decreasing the potential (from 9.48 G for pristine HC700 to 8 G at 0.5 V, and 6.4 G at 0.02 V), in agreement with previous reported works.³³ The changes in the linewidth of the HC700 at different potentials may be explained by alterations in the environment of the free radicals related to the differences in the binding energy of defects/functional groups with Na⁺ ions.^{68, 69} The chemical process reflected in the EPR signals is directly echoed in the electrochemical curves with the presence of a slope, which has been previously explained with the sodiation of the dangling bonds over a range of different energies.^{43, 44}*

Changes to the manuscript (Page 12, line 7).

In general, localised spins obey Curie-Weiss law, whereby the spin susceptibility (which is proportional to the double integration value of the mass-normalised EPR signal) decreases with increasing temperature; and delocalised electrons will show Pauli behaviour, where the spin susceptibility is independent of the temperature.

6) Results presented in the Figure 6 a and b are reduplicated with the results in Figure 2 and 3.

We assume that Reviewer 1 is referring to Figure 5 from the initial manuscript submission as there was no Figure 6 in the submitted manuscript (?). Figures 5a and b have been now removed from the manuscript.

7) The language should be checked again since there are some typos. i.e., in the line 150 on page 5, “in HC materials synthesised.⁷¹ (B187) to 2.01 from other biowaste precursors.”; in the line 397 on page 16, “...towards Curie-Weiss behaviour It is therefore not...”; in Figure 5b, the amplification coefficient of the signal intensity for pristine B187 seems not “x50” as the original signal of B187 is much higher than B1810.

Electron Paramagnetic Resonance Spectroscopy as a Fast Diagnostic Tool to Determine the Sodium Charge Storage Mechanism of Hard Carbon Anodes

The manuscript has been proofread several times by different co-authors and checked against any typos before re-submitting it. The suggested typos were corrected.

Figure 5b has been removed as suggested by the same reviewer in the comment above.

Reviewer #2: EPR is a well-known powerful technique for the characterization of carbon materials. Particularly, the cation storage properties have been the topic of many studies connected with their applicability in different metal-ion batteries. In this context, the manuscript by Bin Wang et al. describes a systematic approach to gaining valuable information from EPR data. The analysis focuses on two HC samples obtained from biowaste and thermally treated at 1000 °C and 700 °C and then subjected to two different milling times and speeds. The microscopic, XRD, gas adsorption, Raman, and XPS characterizations reveal significant differences between samples that are nicely correlated with their EPR spectra at different temperatures and the electrochemical performance in Na half-cells. From the ex-situ EPR data of electrodes at different states of charge, the authors find a clear correlation between the EPR spectral components and the mechanism of sodium storage in two different regions of the cell discharge curve. These conclusions are sound and represent a significant advance in the possibilities of the EPR technique to predict the electrochemical performance of HC electrodes. The publication of the manuscript in its present form is strongly recommended.

We thank Reviewer 2 for their positive comments.

Reviewers' comments:

Reviewer #1 (Remarks to the Author):

The responses are OK and the revisions satisfied. The current version can be accepted.

Reviewer #3 (Remarks to the Author):

In the review entitled "Electron Paramagnetic Resonance spectroscopy as a fast diagnostic tool to determine the sodium charge storage mechanism of hard carbon anodes", the authors propose to investigate the sodium charge storage mechanism of hard carbon electrodes using the electron paramagnetic resonance spectroscopy. To my point of view, such an EPR study can be of interest for battery community. However, I do not see the manuscript fit for publication in Nature Communications because the data analysis is superficial and sometimes incorrect.

Comment 1: Could the authors check the main text again because there are some typos. i.e. in the line 210 on page 8: " HC1000 exhibited a first reversible... (figure3c, tableS3)" this figure represents the results about HC700 sample whereas the authors indicate the HC1000 sample in the main text. In the line 212 on page 8: "while HC700 ... which was slope dominated (figure3c and d)" figure3d shows the result of the HC1000 sample and not the result of the HC700 sample. In the line 213 on page 8 "... from the 1st charge cycle (figure 3e and f)", these both figures do not correspond to the HC700 sample. In the line 324, on page 12, "Post-mortem EPR spectra are shown in figures 7a and c", replace c by b, figure 7c representing the temperature dependence behaviour of simulated EPR data of HC700 and not the EPR signatures. In line 418, on page 15, "This contrasts with the two signals observed in HC100-400-2h...", replace 100 by 1000.

Comment2: Continuous wave EPR signal are typically recorded in the first harmonic detection scheme, as it is also done in the present manuscript. In the supporting information, methods section, the authors present an equation used to simulate the EPR lineshape. Although the function is correct, the authors claimed that if $a=1$, a pure Lorentzian line is obtained. I disagree with this result. In fact, the term $2w.(xc-x)/((xc-x)^2+w^2)^2$ represents the absorption contribution whereas the term $(w^2-(xc-x)^2)/((xc-x)^2+w^2)^2$ is the dispersion part. As a consequence, a pure Lorentzian is obtained only if $a=0$. In contrary, if $a > 0$, a Dysonian-shaped EPR line is obtained. The authors need to clarify that.

Comment 3: On page 11, lines 294-296, the authors claimed that the resonance field (and therefore the g-factor) found for each samples is typical for conduction electrons in carbonaceous materials. However, I very much doubt this interpretation. Such conduction electrons are generally found in graphite materials and not in all carbonaceous materials. Could these signals correspond to carbonaceous radicals ?

Comment 4: On page 12, lines 315-316, the authors found that the EPR signals are centered to a g-value of 2.0024 and state the "... due to the presence of heteroatoms such as oxygen and nitrogen". I very much doubt this interpretation. Classically, a g-value of 2.0034-2.0039 is found for carbon-centered radicals in a nearby oxygen heteroatom and a g-value of 2.01 for oxygen-centered radicals. Why is the assumption of such heteroatoms? Could the authors provide pulsed EPR results such as HYSCORE measurements allowing to probe the local environment of the EPR signals?

Comment 5: On page 12, lines 313-316, the authors claimed that a g-value of 2.0024 (i.e. close to the g-value for free electron named g_e) is indicative of large spin-orbit coupling. This interpretation is wrong. In EPR spectroscopy, a large spin-orbit coupling lead to a g-factor shift from 2.0023. Typically, $\Delta g = g - g_e$ is higher when the orbit and spin magnetic moment are coupled. Here the g-value is very close to 2.0023 indicating a very weak spin-orbit interaction.

Comment 6: On page 12, line 313-314, the authors indicate the presence of "intense and sharp signal" with a g-value of 2.0024 and a peak-to-peak linewidth of 9.5G for the pristine HC700 sample recorded at 10K. However, these both values appear to be found for the broad signal as indicated in the Table S5. Could the authors clarify these values?

Comment 7: On page 12, line 329, the authors relate the paper (ref.33) "Lu, H., et al. Exploring sodium-ion storage mechanism in hard carbons with different microstructure prepared by ball-milling method. Small 14, 1802694 (2018)." However, this paper does not present EPR results whereas the authors claim in the EPR section "A progressive line narrowing ... in agreement with previous reported works [33]".

Comment 8: On page 13, line 359, the authors present the EPR characteristic of a broad signal. It would be better if the authors can specify the sample presented here, to aid the comprehension of experimental data analysis.

Electron Paramagnetic Resonance Spectroscopy as a Fast Diagnostic Tool to Determine the Sodium Charge Storage Mechanism of Hard Carbon Anodes

Point-by-point responses to reviewers:

We are very grateful to Reviewer #3 for their time and careful consideration of our manuscript. We appreciate their advice in helping us deliver the best possible version of the manuscript. Please find our point-by-point responses to their comments.

All changes made to the manuscript and supporting information are highlighted in yellow in the respective documents themselves.

Reviewer #3 (Remarks to the Author):

In the review entitled "Electron Paramagnetic Resonance spectroscopy as a fast diagnostic tool to determine the sodium charge storage mechanism of hard carbon anodes", the authors propose to investigate the sodium charge storage mechanism of hard carbon electrodes using the electron paramagnetic resonance spectroscopy. To my point of view, such an EPR study can be of interest for battery community. However, I do not see the manuscript fit for publication in Nature Communications because the data analysis is superficial and sometimes incorrect.

Comment 1: Could the authors check the main text again because there are some typos. i.e. in the line 210 on page 8: " HC1000 exhibited a first reversible... (figure3c, tableS3)" this figure represents the results about HC700 sample whereas the authors indicate the HC1000 sample in the main text.

We thank Reviewer #3 for highlighting these typos, which mainly arise from the inclusion of new figures in the previous revised version of the manuscript.

Changes to the manuscript:

Before revision (Line 210, page 8)
" HC1000 exhibited a first reversible... (Figure 3c, Table S3)"
After revision
" HC1000 exhibited a first reversible... (Figure 3e and f , Table S3)"

Electron Paramagnetic Resonance Spectroscopy as a Fast Diagnostic Tool to Determine the Sodium Charge Storage Mechanism of Hard Carbon Anodes

In the line 212 on page 8: “while HC700 ... which was slope dominated (figure3c and d)”
figure3d shows the result of the HC1000 sample and not the result of the HC700 sample.

Before revision (Line 212, page 8)
“while HC700 ... which was slope dominated (figure3c and d)”
After revision
“while HC700..., which was slope dominated (Figures 3b and c)

In the line 213 on page 8 “... from the 1st charge cycle (figure 3e and f)”, these both figures do not correspond to the HC700 sample.

Before revision (Line 213, page 8)
“... from the 1st charge cycle (figure 3e and f)”
After revision
“... from the 1st charge cycle (figure 3c and f)”

In the line 324, on page 12, “Post-mortem EPR spectra are shown in figures 7a and c”,
replace c by b, figure 7c representing the temperature dependence behaviour of simulated EPR data of HC700 and not the EPR signatures.

Before revision (Line 324, page 12)
“Post-mortem EPR spectra are shown in figures 7a and c”
After revision
“Post-mortem EPR spectra are shown in Figures 7a and b ”

Electron Paramagnetic Resonance Spectroscopy as a Fast Diagnostic Tool to Determine the Sodium Charge Storage Mechanism of Hard Carbon Anodes

In line 418, on page 15, “This contrasts with the two signals observed in HC100-400-2h...”, replace 100 by 1000.

Before revision
“This contrasts with the two signals observed in HC100-400-2h...”
After revision
“This contrasts with the two signals observed in HC1000-400-2h...”

Comment2: Continuous wave EPR signal are typically recorded in the first harmonic detection scheme, as it is also done in the present manuscript.

In the supporting information, methods section, the authors present an equation used to simulate the EPR lineshape.

Although the function is correct, the authors claimed that if $a=1$, a pure Lorentzian line is obtained.

I disagree with this result. In fact, the term $2w.(xc-x)/((xc-x)^2+w^2)^2$ represents the absorption contribution whereas the term $(w^2-(xc-x)^2)/((xc-x)^2+w^2)^2$ is the dispersion part. As a consequence, a pure Lorentzian is obtained only if $a=0$. In contrary, if $a > 0$, a Dysonian-shaped EPR line is obtained. The authors need to clarify that.

We thank Reviewer #3 for highlighting this and agree with them on their statement that if $a=0$ in Equation S7, a Lorentzian line is obtained and if $a > 0$ a Dysonian-shape line is obtained. Our statement comes as a result of a typo in the experimental description provided in the method section of the SI and has not affected any of the results shown in the manuscript. We have now corrected the text accordingly.

Changes to the manuscript:

Before revision (Methods section, Supplementary Information)
The EPR lineshape simulation of a single component was performed in MATLAB ¹ using the following equation: ⁹

Electron Paramagnetic Resonance Spectroscopy as a Fast Diagnostic Tool to Determine the Sodium Charge Storage Mechanism of Hard Carbon Anodes

$$y = y_0 + C \times \frac{2w \times (x_c - x)}{((x_c - x)^2 + w^2)^2} + C \times a \times \frac{w^2 - (x_c - x)^2}{((x_c - x)^2 + w^2)^2} \quad (\text{Equation S7})$$

where y is the signal intensity, y_0 is the background, C is the amplitude factor, x is the magnetic field, x_c is the centre magnetic field of the EPR signal, w is the linewidth, a is the ratio of the dispersion to absorption with a range from 0 to 1. If $a = 1$, the EPR signal is the symmetric Lorentzian lineshape. If $a \neq 1$, a Dysonian lineshape is considered.

After revision

The EPR lineshape simulation of a single component was performed in MATLAB¹ using **the 1st derivative of the Dysonian equation reported in the literature:⁹**

$$y = y_0 + C \times \frac{2w \times (x_c - x)}{((x_c - x)^2 + w^2)^2} + C \times a \times \frac{w^2 - (x_c - x)^2}{((x_c - x)^2 + w^2)^2} \quad (\text{Equation S7})$$

where y is the signal intensity, y_0 is the background, C is the amplitude factor, x is the magnetic field, x_c is the centre magnetic field of the EPR signal, w is the linewidth, a is the ratio of the dispersion to absorption with a range from 0 to 1. **If $a = 0$** , the EPR signal is the symmetric Lorentzian lineshape. **If $a > 0$** , a Dysonian lineshape is considered. The Dysonian lineshape, with its characteristic asymmetric factor A/B , corresponds to the ratio of intensities of the spectral positive peak to the negative one.

Comment 3: On page 11, lines 294-296, the authors claimed that the resonance field (and therefore the g-factor) found for each sample is typical for conduction electrons in carbonaceous materials. However, I very much doubt this interpretation. Such conduction electrons are generally found in graphite materials and not in all carbonaceous materials. Could these signals correspond to carbonaceous radicals?

We concur with the reviewers' comments and appreciate that they raised this point so we can clarify the text in our manuscript.

Our sentence on line 294, page 11 was, unfortunately, inaccurate, as we should have referred to graphite-like materials and not using such a broad term as "carbonaceous materials". Similarly, referring to only "conduction electrons" was incorrect as these signals can also correspond to carbonaceous radicals. Indeed, in our investigation we showed how signals arising from conduction electrons and radicals will give rise to different behaviour with regards to temperature dependence results, in accordance with Pauli and the Curie-Weiss laws. This

Electron Paramagnetic Resonance Spectroscopy as a Fast Diagnostic Tool to Determine the Sodium Charge Storage Mechanism of Hard Carbon Anodes

rigorous classification method has demonstrated an excellent correspondence with the plateau and slope capacities observed in hard carbon materials when used in sodium-ion batteries.

We have amended the text accordingly based on these suggestions.

Changes to the manuscript:

Before revision
All the EPR signals are observed at resonance fields which are typical for conduction electrons in carbonaceous materials. Therefore, we can exclude any contribution from paramagnetic species arising from impurities from the ball milling reactions, which would be characterised by different g tensor values.
After revision
All the EPR signals are observed at resonance fields which are typical for carbonaceous radicals and conduction electrons in graphite-like domains as described below. Therefore, we can exclude any contribution from paramagnetic species arising from impurities from the ball milling reactions, which would be characterised by different g tensor values.

Comment 4: On page 12, lines 315-316, the authors found that the EPR signals are centered to a g-value of 2.0024 and state the "... due to the presence of heteroatoms such as oxygen and nitrogen". I very much doubt this interpretation.

Classically, a g-value of 2.0034-2.0039 is found for carbon-centered radicals in a nearby oxygen heteroatom and a g-value of 2.01 for oxygen-centered radicals.

Why is the assumption of such heteroatoms?

We have provided evidence in the manuscript for the presence of heteroatoms such as N and O such as Raman and X-ray photoelectron spectroscopy, where we showed that for HC700, the C/O ratio was 25.6. Our intention with this sentence was not to claim that we had oxygen or nitrogen centred radicals in this sample but that these heteroatoms might have an effect on the g value observed for this sample. We agree with Reviewer 3 that based on the g values obtained, these correspond to carbon-centred radicals and have amended the text accordingly

Electron Paramagnetic Resonance Spectroscopy as a Fast Diagnostic Tool to Determine the Sodium Charge Storage Mechanism of Hard Carbon Anodes

to clarify this. We have found similar statements in the literature regarding graphene, which reported g factor values of ca. 2.0023 and attributed these to the presence of functional groups (*Nanomaterials* **2020**, *10*(4), 798 and *J. Mater. Chem. C*, 2014,**2**, 8105-8112 and A. Barbon, in *Electron Paramagnetic Resonance: Volume 26*, ed. V. Chechik and D. M. Murphy, The Royal Society of Chemistry, 2018, vol. 26, pp. 38-65).

Changes to the manuscript:

Before revision
EPR spectrum of pristine HC700 at 10 K showed an intense and sharp signal with a Lorentzian lineshape centred at $g = 2.0024$ and $\Delta H_{pp} = 9.5$ G (Figure 5a) that obeyed the Curie-Weiss law, as expected from localised spin centres (Figure 6a).³⁸ The spin g-factor is close to the g-factor for a free electron ($g_e = 2.0023$), which indicates that the sample has a large spin-orbit coupling constant due to the presence of heteroatoms such as oxygen and nitrogen.³⁸ EPR parameters of the signal are consistent with localised radicals, which can be attributed to a defective carbon structure, such as dangling bonds with terminating oxygen/nitrogen groups (as observed in the XPS data) in the edge of the graphene sheets and/or on the surface of open micropores and vacancies within the graphene sheets.^{75, 76}
After revision
EPR spectrum of pristine HC700 at 10 K showed an intense and sharp signal with a Lorentzian lineshape centred at $g = 2.0024$ and $\Delta H_{pp} = 9.5$ G (Figure 5a) that obeyed the Curie-Weiss law, as expected from localised spin centres (Figure 6a).³⁸ EPR parameters of the signal are consistent with localised carbon centred radicals, which can be attributed to a defective carbon structure, such as dangling bonds with terminating oxygen/nitrogen groups (as observed in the XPS data) in the edge of the graphene sheets and/or on the surface of open micropores and vacancies within the graphene sheets.^{75, 76}

Could the authors provide pulsed EPR results such as HYSCORE measurements allowing to probe the local environment of the EPR signals?

Electron Paramagnetic Resonance Spectroscopy as a Fast Diagnostic Tool to Determine the Sodium Charge Storage Mechanism of Hard Carbon Anodes

This presents a commendable perspective. Indeed, relevant scholars have already delved into pulsed EPR studies of carbon materials, and our previous endeavours have involved similar experimental work. However, it is worth noting that carbon materials inherently exhibit relatively weak signals in EPR, posing significant experimental challenges. Remarkably, in the domain of energy storage carbon materials, the utilization of EPR as a primary investigative approach is not yet widely prevalent.

In this manuscript, our primary objective is to showcase the simplicity and efficacy of utilizing room-temperature X-band EPR testing to discern the sodium storage type in hard carbon negative electrode materials, effectively transforming EPR spectra into a diagnostic tool. Undoubtedly, your insightful suggestion, particularly emphasizing the implementation of pulsed EPR to finely elucidate the carbon electron environment, energy storage mechanisms, and structural evolution of carbon materials, would present an engrossing avenue for further exploration. Such investigations promise to yield a captivating body of work, illuminating essential aspects of carbon-based energy storage materials but we believe they fall out of the remit of this manuscript.

Comment 5: On page 12, lines 313-316, the authors claimed that a g -value of 2.0024 (i.e. close to the g -value for free electron named g_e) is indicative of large spin-orbit coupling. This interpretation is wrong. In EPR spectroscopy, a large spin-orbit coupling lead to a g -factor shift from 2.0023. Typically, $\Delta g = g - g_e$ is higher when the orbit and spin magnetic moment are coupled. Here the g -value is very close to 2.0023 indicating a very weak spin-orbit interaction. We would like to refer the reviewer to our response to comment 4. Additionally, we understand the concerns from the reviewer regarding our “large” spin-orbit wording in the manuscript and we have amended the text in the manuscript accordingly. We used the term “large” to indicate that the g value had changed from 2.0023 but we accept that this was a poor word choice since the shift is quite insignificant.

This change made in the manuscript should not affect any of the results shown in the manuscript and should have not affected any of the discussions provided in the manuscript.

Changes to the manuscript:

Before revision (Line 313, page 12)
EPR spectrum of pristine HC700 at 10 K showed an intense and sharp signal with a Lorentzian lineshape centred at $g = 2.0024$ and $\Delta H_{pp} = 9.5$ G (Figure 5a) that obeyed the Curie-Weiss law, as expected from localised spin centres (Figure 6a). ³⁸ The spin g -factor

Electron Paramagnetic Resonance Spectroscopy as a Fast Diagnostic Tool to Determine the Sodium Charge Storage Mechanism of Hard Carbon Anodes

is close to the g -factor for a free electron ($g_e = 2.0023$), which indicates that the sample has a large spin-orbit coupling constant due to the presence of heteroatoms such as oxygen and nitrogen.³⁸

EPR parameters....

After revision

EPR spectrum of pristine HC700 at 10 K showed an intense and sharp signal with a Lorentzian lineshape centred at $g = 2.0024$ and $\Delta H_{pp} = 9.5$ G (**Figure 5a**) that obeyed the Curie-Weiss law, as expected from localised spin centres (**Figure 6a**).³⁸

EPR parameters.....

Comment 6: On page 12, line 313-314, the authors indicate the presence of “intense and sharp signal” with a g -value of 2.0024 and a peak-to-peak linewidth of 9.5G for the pristine HC700 sample recorded at 10K. However, these both values appear to be found for the broad signal as indicated in the Table S5. Could the authors clarify these values?

Although the g , $\Delta H_{pp}/G$ and lineshape information provided for HC700 are correct in Tables S5 (RT) and S6 (10K), we understand that Reviewer 3 might have found confusing finding this information under the “Broad signal” column.

Our intention was to highlight that this signal was broader than any of the narrow signals arising from the ball milled samples but it is indeed not a broad signal, if referring to the ΔH_{pp} value obtained. Nevertheless, we understand that without further explanation, this was ambiguous.

As a result, we believe that it is more comprehensive to separate our table in two sections consisting of pristine and ball-milled samples. In this way, we do not require any signal “classification” for HC700, and indeed from the ΔH_{pp} value one can see that this corresponds to a narrow signal, which is indeed narrower than that observed in HC1000.

Furthermore, although not requested by the Reviewer, we have also modified based on the above comment Figure 6 accordingly, where the HC700 signal was categorised as a broad signal.

Changes to the SI:

Electron Paramagnetic Resonance Spectroscopy as a Fast Diagnostic Tool to Determine the Sodium Charge Storage Mechanism of Hard Carbon Anodes

Before revision						
Table S5						
Sample@ RT	Broad signal			Narrow signal		
	g	$\Delta H_{pp}/G$	Lineshape	g	$\Delta H_{pp}/G$	Lineshape
HC700	2.0025	32	L*	-	-	
HC700-400-2h	2.0021	54	L*	2.0023	5.8	L*
HC700-400-5h	2.0024	118	L*	2.0031	10.3	L*
HC1000	2.0033	210	D (1.12) **	-	-	-
HC1000-400-2h	2.0029	200	D (1.01) **	2.0026	6.5	L*
HC1000-400-5h	2.0029	27.7	D (1.1) **	2.0029	6.4	L*
Table S6						
Sample@ 10K	Broad signal			Narrow signal		
	g	$\Delta H_{pp}/G$	Lineshape	g	$\Delta H_{pp}/G$	Lineshape
HC700	2.0024	9.5	L*	-	-	
HC700-400- 2h	2.0026	26.5	L*	2.0036	4.9	L*
HC700-400- 5h	2.0025	72.9	L*	2.0026	9.12	L*
HC1000	2.0044	150	D (1.07) **	-	-	-
HC1000-400- 2h	2.0025	140	D (1.03) **	2.003	6.1	L*
HC1000-400- 5h	2.0027	24.9	D (1.04) **	2.0034	6.9	L*
After revision						
Table S5.						
Pristine samples @ RT	g	$\Delta H_{pp}/$ G	Lineshap e			
HC700	2.002 5	32	L*			

Electron Paramagnetic Resonance Spectroscopy as a Fast Diagnostic Tool to Determine the Sodium Charge Storage Mechanism of Hard Carbon Anodes

HC1000	2.003 3	210	D (1.12) **			
Ball milled samples @ RT						
		Broad signal		Narrow signal		
	g	$\Delta H_{pp}/G$	Lineshap e	g	$\Delta H_{pp}/G$	Lineshape
HC700-400-2h	2.002 1	54	L*	2.0023	5.8	L*
HC700-400-5h	2.002 4	118	L*	2.0031	10.3	L*
HC1000-400-2h	2.002 9	200	D (1.01) **	2.0026	6.5	L*
HC1000-400-5h	2.002 9	27.7	D (1.1) **	2.0029	6.4	L*
Table S6.						
Pristine samples @ 10K						
	g	$\Delta H_{pp}/G$	Lineshap e			
HC700	2.002 4	9.5	L*			
HC1000	2.004 4	150	D (1.07) **			
Ball milled samples @ 10K						
		Broad signal		Narrow signal		
	g	$\Delta H_{pp}/G$	Lineshap e	g	$\Delta H_{pp}/G$	Lineshap e
HC700-400-2h	2.002 6	26.5	L*	2.0036	4.9	L*
HC700-400-5h	2.002 5	72.9	L*	2.0026	9.12	L*
HC1000-400-2h	2.002 5	140	D (1.03) **	2.003	6.1	L*

Electron Paramagnetic Resonance Spectroscopy as a Fast Diagnostic Tool to Determine the Sodium Charge Storage Mechanism of Hard Carbon Anodes

HC1000-400-5h	2.002 7	24.9	D (1.04) **	2.0034	6.9	L*
---------------	------------	------	----------------	--------	-----	----

Changes to the manuscript:

Electron Paramagnetic Resonance Spectroscopy as a Fast Diagnostic Tool to Determine the Sodium Charge Storage Mechanism of Hard Carbon Anodes

Figure 6. Temperature dependence behaviour of the (a) pristine material, (b, d) broad and (c, e) narrow contributions obtained by lineshape simulation for all the ~~pristine and~~ ball-milled HC700 and HC1000 samples. The double integration (DI) value of the EPR signal is proportional to the spin density and is calculated from the absolute area value of the EPR signal. The solid line in the spectra shows the Curie-Weiss fitting results. (f) Normalised DI value (refers to the spin susceptibility) $\times T$ vs. T of the simulated signals of HC700 and HC100 pristine materials and broad signals of the ball milled samples and shown in Figures 6a, b and d.

Comment 7: On page 12, line 329, the authors relate the paper (ref.33) “Lu, H., et al. Exploring sodium-ion storage mechanism in hard carbons with different microstructure prepared by ball-milling method. Small 14, 1802694 (2018).” However, this paper does not present EPR results whereas the authors claim in the EPR section “A progressive line narrowing ... in agreement with previous reported works [33]”.

We apologise for this typo- The correct reference should be ref 44., which was already in our manuscript. We have amended this reference in the main text accordingly.

The ex situ EPR test results in Figure 4b of this article (Small Methods 5, 2100580 (2021), ref 44) reveal a process of linewidth narrowing, although the authors did not provide an in-

Electron Paramagnetic Resonance Spectroscopy as a Fast Diagnostic Tool to Determine the Sodium Charge Storage Mechanism of Hard Carbon Anodes

depth discussion on this matter. Interestingly, similar results were observed in another article authored by the same researchers (J. Am. Chem. Soc. 2019, 141, 24, 9623–9628), where they use ex situ EPR tests. These findings suggest that the materials under investigation primarily involve Na ion adsorption processes, which aligns with the focus of the HC700 study.

Changes to the manuscript:

Before revision (Line 329, page 12)
“A progressive line narrowing ... in agreement with previous reported works [33]”.
After revision
“A progressive line narrowing ... in agreement with previous reported works [44]”.

Comment 8: On page 13, line 359, the authors present the EPR characteristic of a broad signal. It would be better if the authors can specify the sample presented here, to aid the comprehension of experimental data analysis.

We have included the sample name (HC1000) at the start of the sentence “A broad signal...” to clarify the sample that we are referring to in our text.

Changes to the manuscript:

Before revision (Line 359, Page 13)
A broad signal with an asymmetric (Dysonian) lineshape ($A/B= 1.07$) ($g = 2.0044$ and $\Delta H_{pp} \approx 150$ G).....
After revision
HC1000 showed a broad signal with an asymmetric (Dysonian) lineshape ($A/B= 1.07$) ($g = 2.0044$ and $\Delta H_{pp} \approx 150$ G).....

REVIEWERS' COMMENTS

Reviewer #3 (Remarks to the Author):

The clarifications regarding the interpretations are satisfied. The manuscript can be published in its present form.